# LEAKAGE AND SECOND-ORDER DYNAMICS IMPROVE HIPPOCAMPAL RNN REPLAY

## ABSTRACT

Biological neural networks (like the hippocampus) can internally generate "replay" resembling stimulus-driven activity. Recent computational models of replay use noisy recurrent neural networks (RNNs) trained to path-integrate. Replay in these networks has been described as Langevin sampling, but new modifiers of noisy RNN replay have surpassed this description. We re-examine noisy RNN replay as sampling to understand or improve it in three ways: (1) Under simple assumptions, we prove that the gradients replay activity should follow are time-varying and difficult to estimate, but readily motivate the use of hidden state leakage in RNNs for replay. (2) We confirm that hidden state adaptation (negative feedback) encourages exploration in replay, but show that it incurs non-Markov sampling that also slows replay. (3) We propose the first model of temporally compressed replay in noisy path-integrating RNNs through hidden state momentum, connect it to underdamped Langevin sampling, and show that, when combined with adaptation, it counters slowness while maintaining exploration. We verify our findings via path-integration of 2D paths in T-maze and triangular environments and of high-dimensional paths of synthetic rat place cell activity.

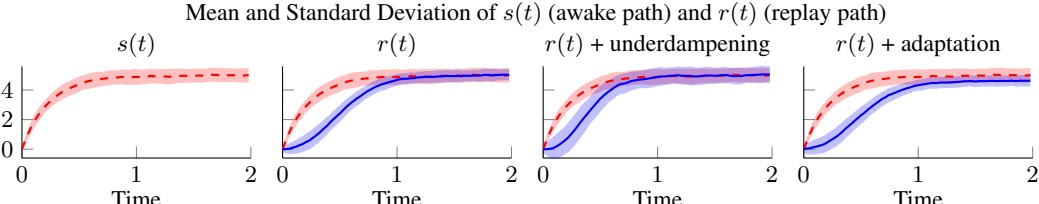

Figure 1: **Underdamped dynamics accelerate offline replay, adaptation slows it.** Here we simulate a noisy RNN $r(t)$ that optimally path-integrates an Ornstein-Uhlenbeck process $s(t)$ from its velocity $s'(t)$. We assume $r(t)$ minimizes the loss in Equation (8) and thus evolves according to its *score function* $\nabla_{r(t)} \log p(r(t))$ (Equations 12 and 16), performing a variant of Langevin sampling when no input is given. Above, we compare three modifiers of RNN activity: the default (no modification, a.k.a. *overdamped*), our proposed *underdamped* (momentum), and *adaptation* (negative feedback) dynamics. Each modifier affects the replay distribution $p(r(t))$ in different ways: underdamped sampling accelerates $p(r(t))$ towards $p(s(t))$, decreasing the distance between them, while adaptation slows convergence of $p(r(t))$ towards $p(s(t))$, increasing this distance.

## 1 INTRODUCTION

During quiescent periods such as sleep or wakeful resting, some neural circuits internally generate activity resembling that of active periods (Tingley & Peyrache, 2020). Such "replay" phenomena have been observed in the prefontal (Euston et al., 2007; Peyrache et al., 2009), sensory (Kenet et al., 2003; Xu et al., 2012), motor (Hoffman & McNaughton, 2002), and entorhinal cortices (Gardner et al., 2022); the anterior thalamus (Peyrache et al., 2015); and the hippocampus (Buzsáki, 1986; Skaggs & McNaughton, 1996; Nádasdy et al., 1999; Lee & Wilson, 2002; Foster, 2017). Of these circuits, the hippocampus is particularly interesting because its robustness in tasks like navigation (O'Keefe & Nadel, 1978; Burgess et al., 1994; McNaughton et al., 1996) and planning (Pfeiffer &

Foster, 2013; Miller et al., 2017) during active states seems crucially tied to its spontaneous activity during quiescent states (Buzsáki, 1989; 2015; Tononi & Cirelli, 2014; Ólafsdóttir et al., 2015; 2018).

While some works have produced replay using supervised generative models (Deperrois et al., 2022), most existing models of hippocampal activity treat replay as an emergent byproduct of careful network design. Relevant network parameters include connectivity structures (Shen & McNaughton, 1996; Milstein et al., 2023), local plasticity mechanisms (Hopfield, 2010; Litwin-Kumar & Doiron, 2014; Theodoni et al., 2018; Haga & Fukai, 2018; Asabuki & Fukai, 2025), firing rate adaptation (Chu et al., 2024; Azizi et al., 2013; Itskov et al., 2011; Dong et al., 2021; Li et al., 2024), and input modulation (Kang & DeWeese, 2019). While these models reproduce aspects of replay, they are typically motivated by empirical findings and lack rigorous theoretical justification.

A more principled model of hippocampal function with emergent replay is sequential predictive learning (Krishna et al., 2024; Levenstein et al., 2024), wherein neural circuits predict dynamic environment or task variables from imperfect observations thereof, e.g., path-integrating velocity measurements to track a position. This normative description of the hippocampus as a sequence predictor (Levy, 1989; Stachenfeld et al., 2017) matches hippocampal encodings of upcoming stimuli (Davachi & DuBrow, 2015) and prediction errors (Aitken & Kok, 2022; Miller et al., 2023), and neural activity sweeps that represent possible future trajectories (Kay et al., 2020; Johnson & Redish, 2007). Unlike traditional, hand-crafted models of hippocampal circuits (i.e., continuous attractor networks (Samsonovich & McNaughton, 1997; Battaglia & Treves, 1998)), sequential predictive learning models are trained from data. Nonetheless, they account for the emergence of place cells (Recanatesi et al., 2021; Levenstein et al., 2024; Chen et al., 2024), grid cells (Cueva & Wei, 2018; Sorscher et al., 2019), and head direction cells (Cueva et al., 2020; Uria et al., 2022); can incorporate phenomena like theta oscillations (Levenstein et al., 2024); and exhibit quiescent replay activity (Krishna et al., 2024; Levenstein et al., 2024; Chen et al., 2024).

Krishna et al. (2024) provided the first theoretical foundation for replay in sequential predictive learning networks, showing analytically that they generate diffusive replay (Stella et al., 2019) during quiescent activity (i.e., noise-driven activity in the absence of inputs) by Langevin sampling (Besag, 1994) from the waking activity distribution using its score function. Subsequent empirical work (Levenstein et al., 2024) introduced new mechanisms to induce *exploration* in replay through negative feedback, i.e., neural adaptation. Exploration is the notion that replay expresses a variety of behavioral sequences (Davidson et al., 2009; Pfeiffer, 2020), and is associated with long trajectories in neural space, visitation of multiple attractor basins, and transitions that were not present in awake activity. Adaptation can destabilize attractors and induce sudden transitions in replay activity (Itskov et al., 2011; Dong et al., 2021; Li et al., 2024; Levenstein et al., 2024), thereby facilitating exploration, and is thought to play a key role in the dynamics of replay *in vivo* (Levenstein et al., 2019). However, existing theory on sequential predictive learning cannot account for these mechanisms. Furthermore, sequential predictive learning models do not currently account for the *temporal compression* of replay sequences relative to awake sequences of activity (Nádasdy et al., 1999; Buzsáki, 2015; Michelmann et al., 2019; Farooq & Dragoi, 2019). This phenomenon, which could be caused by short-term facilitation (Leibold et al., 2008; Jaramillo & Kempter, 2017), is not currently captured in replay from any trained RNN model to our knowledge.

Overall, while sequential predictive learning is a promising model of hippocampal function and replay, its recent empirical advances have outpaced its theoretical foundations. Moreover, it is unclear how to incorporate phenomena like temporal compression in these models, or what inductive biases in these RNNs' design affect replay. We remedy these shortcomings by characterizing how RNN design and Langevin sampling statistics affect each other. Some results describe how RNN design affects the speed and variance of replay activity, while others start from the Langevin sampling formulation of replay and either explain existing architectural choices as useful inductive biases, or propose new mechanisms to again modulate sampling. In summary, we answer three questions:

1. Optimal path-integration in the presence of noise requires RNNs to learn the score function of the noisy activity distribution. What is this function, might it inform RNN design? *The score function is time-variant and difficult to estimate, even for simple distributions, but our expression of it motivates the addition of leakage (linear dynamics) in RNNs.*

2. Adaptation (negative feedback) empirically induces exploration in replay. How does it affect Langevin sampling? *Adaptation induces non-Markov second-order Langevin sampling that destabilizes attractors, which can both help (diversify) and hurt (slow) replay.*

3. Traditional generative models benefit from a wide array of sampling techniques. Could new sampling methods improve replay in noisy RNNs? *Underdamped Langevin sampling via momentum quickens neural replay, like temporal compression induced by short-term facilitation* in vivo*, and mitigates slowing from adaptation while maintaining exploration.*

## 2 BACKGROUND

In this section, we provide background and summarize results from prior work (Krishna et al., 2024) that has described replay in path-integrating neural circuits as the sampling during quiescence (i.e., the absence of any inputs) of neural activity states from the distribution of waking, task-like activity. In short, the dynamics of recurrent networks that learn to optimally path-integrate noisy inputs cause the network's states even in the absence of inputs to resemble those attained during actual task performance. We also provide details on mechanisms used in the literature to improve or bias replay, which we explore the effects of in more detail in this work. For an overview of this section, see Appendix B.

### 2.1 LANGEVIN DYNAMICS

Langevin sampling from an unknown distribution $p(x)$ entails stochastic gradient ascent of an iterate $x(t)$ along the log-likelihood of $p(x)$ via its *score function* $\nabla \log p(x)$ or an estimate thereof:

$$x'(t) = x(t) + \nabla \log p(x) + \sqrt{2}\eta(t), \qquad (1)$$

where $\eta(t)$ is Gaussian white noise. While Equation 1 describes *overdamped* dynamics, there also exist *underdamped* Langevin dynamics[1] (Equations 2 or 3, see Chapter 6 of Pavliotis (2014)) that converge faster to the target distribution $p(x)$ and better utilize noisy gradients (Cheng et al., 2018):

$$x''(t) = \nabla \log p(x) - \gamma x'(t) + \sqrt{2\gamma}\eta(t), \quad \text{or} \qquad (2)$$

$$x'(t) = v(t), \quad v'(t) = \nabla \log p(x) - \gamma v(t) + \sqrt{2\gamma}\eta(t) \qquad (3)$$

In this work, we consider noisy RNNs that have implicitly learned to perform Langevin sampling of their own, fixed distributions of activity during task performance, when driven by just intrinsic noise and in the absence of inputs. That is, the activity of the RNNs at each unrolled timestep in the absence of inputs represents a plausible and likely vector of network activity during actual task performance in the presence of inputs. In particular, we view such networks in the context of replay, where neural circuits recapitulate task-like activity even during sleep.

### 2.2 OFFLINE REPLAY IN RNNS

This work focuses on RNNs that implicitly learn to act as generative models over a fixed distribution of input sequences. Krishna et al. (2024) have shown how noisy RNNs trained to *path-integrate* their inputs implicitly learn statistics that produce Langevin sampling of their own task-relevant activity distribution when no input is given, demonstrating statistically faithful replay sequences during quiescence. That is, the RNNs' activity in the absence of inputs "replays" states from the same distribution as RNN activity with inputs during actual task performance. This leads to the generation or "replay" of output sequences resembling those that the network sees during training or task performance with inputs. Path-integration is particularly relevant to neuroscience: animals can leverage motion cues, observations, or prior experiences to accurately estimate positions (Seelig & Jayaraman, 2015; Chrastil, 2025), and neural circuits like the entorhinal cortex (Sorscher et al., 2019) have been identified to perform such computations. Here we summarize the finding that noisy RNNs trained to path-integrate input time-series learn the *score function* of the input distribution (Krishna et al., 2024).

---

[1]Discretized underdamped Langevin dynamics are a form of Hamiltonian MCMC (Cheng et al., 2018).

**Definition 1.** A *noisy recurrent neural network (RNN)* has hidden states $\boldsymbol{r}(t)$ that evolve at each timestep $t$ via some (nonlinear) function of its previous hidden states, an input signal $\boldsymbol{u}(t)$, and noise:

$$\boldsymbol{r}(t + \Delta t) = f(\boldsymbol{r}(t), \boldsymbol{u}(t), \sigma_r \boldsymbol{\eta}(t)) \tag{4}$$

**Definition 2.** A *path-integration* objective $\mathcal{L}(t)$ penalizes the difference between a state variable $\boldsymbol{s}(t)$ and a learnable linear projection of the RNN hidden state $\boldsymbol{r}(t)$ at every timestep $t$:

$$\mathcal{L}(t) = \mathbb{E}_{\boldsymbol{\eta}} \|\boldsymbol{s}(t) - \boldsymbol{D}\boldsymbol{r}(t)\|_2 \tag{5}$$

**Assumption 1.** *Krishna et al. (2024) assume the hidden state dynamics of a noisy RNN can be decomposed as the sum of two separate functions[2] and a white noise term $\sigma_r \boldsymbol{\eta}(t) \sim \mathcal{N}(0, \sigma_r^2 \Delta t)$.*

$$\boldsymbol{r}(t + \Delta t) - \boldsymbol{r}(t) = \Delta \boldsymbol{r}(t) \approx \Delta \boldsymbol{r}_1(t) + \Delta \boldsymbol{r}_2(t) + \sigma_r \boldsymbol{\eta}(t) \tag{6}$$

**Assumption 2.** *The optimal $\boldsymbol{r}(t)$ minimizes $\mathcal{L}(t)$ such that $p(\boldsymbol{r}^*(t))$ is normal around $\boldsymbol{D}^\dagger \boldsymbol{s}(t)$:*

$$p(\boldsymbol{r}^*(t)|\boldsymbol{s}(t)) \sim \mathcal{N}(\boldsymbol{D}^\dagger \boldsymbol{s}(t), \boldsymbol{I}\sigma_r^2 \Delta t) \tag{7}$$

**Lemma 1.** *With Assumptions 1 and 2, $\mathcal{L}(t + \Delta t)$ is upper bounded by tracking and denoising terms:*

$$\mathcal{L}(t + \Delta t) \leq \mathcal{L}_{upper}(t + \Delta t) = \|\boldsymbol{s}'(t) - \boldsymbol{D}\Delta \boldsymbol{r}_2(t)\| + \|\boldsymbol{D}\|_F \mathbb{E}_{\boldsymbol{\eta}} \|\Delta \boldsymbol{r}_1(t) + \sigma_r \boldsymbol{\eta}(t - \Delta t)\|_2 \\ + \mathbb{E}_{\boldsymbol{\eta}} \|\boldsymbol{D}\sigma_r \boldsymbol{\eta}(t)\|_2 \tag{8}$$

**Assumption 3.** *The optimal $\boldsymbol{r}(t)$ greedily minimizes $\mathcal{L}_{upper}$ at each time $t$ only.*

$$\{\boldsymbol{r}^*(t)\}_{t=0}^T = \operatorname*{arg\,min}_{\{\boldsymbol{r}(t)\}_{t=0}^T} \int_{t=0}^T \mathcal{L}(t + \Delta t) \Delta t \approx \left\{ \operatorname*{arg\,min}_{\boldsymbol{r}(t)} \mathcal{L}_{upper}(t + \Delta t) \right\}_{t=0}^T \tag{9}$$

**Theorem 1.** *Given Assumption 3, the optimal $\Delta \boldsymbol{r}(t)$ follows $\boldsymbol{s}'(t)$ and the score function of $p(\boldsymbol{r}(t))$.*

$$\Delta \boldsymbol{r}^*(t) = \operatorname*{arg\,min}_{\Delta \boldsymbol{r}_2(t)} \mathcal{L}_{upper}(t + \Delta t) + \operatorname*{arg\,min}_{\Delta \boldsymbol{r}_1(t)} \mathcal{L}_{upper}(t + \Delta t) + \sigma_r \boldsymbol{\eta}(t) \tag{10}$$

$$= \boldsymbol{D}^\dagger \boldsymbol{s}'(t) \Delta t + \sigma_r^2 \Delta t \nabla_{\boldsymbol{r}(t)} \log p(\boldsymbol{r}(t)) + \sigma_r \boldsymbol{\eta}(t) \tag{11}$$

During training, noisy RNNs are (unless otherwise stated, see Section 2.3) provided $\boldsymbol{u}(t) = \boldsymbol{s}'(t)$ (or a nonlinear observation thereof) to perform path-integration and focus on denoising.

**Theorem 2.** *In the absence of input (quiescence), a noisy RNN already trained to path-integrate $\boldsymbol{s}(t)$ from $\boldsymbol{s}'(t)$ will perform gradient ascent along the score function of $\boldsymbol{r}(t)$. If $p(\boldsymbol{r}(t))$ and $p(\boldsymbol{s}(t))$ are stationary, then this ascent is Langevin sampling (Equation 1). If the variance of $\sigma_r \boldsymbol{\eta}(t)$ is scaled by a factor of 2, then $p(\boldsymbol{r}(t))$ is guaranteed to converge to the steady-state distribution $p(\boldsymbol{r}) = p(\boldsymbol{D}^\dagger \boldsymbol{s})$.*

$$\text{If } \boldsymbol{r}(t + \Delta t) = \boldsymbol{r}(t) + \sigma_r^2 \Delta t \nabla_{\boldsymbol{r}(t)} \log p(\boldsymbol{r}(t)) + \sqrt{2}\sigma_r \boldsymbol{\eta}(t), \text{ then } \lim_{t \to \infty} p(\boldsymbol{r}(t)) = p(\boldsymbol{D}^\dagger \boldsymbol{s}) \tag{12}$$

## 2.3 Existing Methods of Biasing Replay in RNNs

**Neural adaptation.** Biological neurons can mitigate prolonged or low-frequency activity via negative feedback, or *adaptation* (Benda, 2021; Gutkin & Zeldenrust, 2014). This feedback has proven important for describing *in vivo* hippocampal activity and replay (Itskov et al., 2011; Levenstein et al., 2019), and in computational models of replay has been shown to encourage long replay trajectories (exploration) by preventing neural activations from getting stuck in attractor basins (Dong et al., 2021; Li et al., 2024; Levenstein et al., 2024). Like Levenstein et al. (2024), we define adaptation as negative feedback $\boldsymbol{c}(t)$ added to RNN activity $\boldsymbol{r}(t)$ (Equation 4) after training[3]:

$$\boldsymbol{r}(t + \Delta t) = f(\boldsymbol{r}(t), \boldsymbol{u}(t), \sigma_r \boldsymbol{\eta}(t)) - \boldsymbol{c}(t), \quad \Delta \boldsymbol{c}(t) = \frac{1}{\tau_a}(-\boldsymbol{c}(t) + b_a \boldsymbol{r}(t)) \tag{13}$$

---

[2]Each function $\Delta \boldsymbol{r}_1(t), \Delta \boldsymbol{r}_2(t)$ can depend on variables beyond $t$, but they are omitted for concision.

[3]Subtraction of the moving average $\boldsymbol{c}(t)$ also arises naturally from greedy minimization of $\mathcal{L}(t) + \frac{1}{2}\|\boldsymbol{c}(t)\|_2^2$.

**Masked training.** Denoisers and autoencoders benefit from *masked training*, wherein some regions of input data are set to zero before model processing (Zhang et al., 2023). Levenstein et al. (2024) introduce masked training for path-integration by periodically masking the input $\boldsymbol{u}(t)$ (the observation of $\boldsymbol{s}'(t)$): only at every $k$-th timestep does the RNN observe a nonzero input (Equation 14).

$$\boldsymbol{u}(t) = \begin{cases} \boldsymbol{s}'(t), & t \bmod k = 0 \\ \boldsymbol{0}, & \text{otherwise} \end{cases} \tag{14}$$

Levenstein et al. (2024) found that masked training makes replay sequences more coherent and makes manifolds of neural activity more similar to the spatial layout of the environment. We found that masked training improves replay stability, so we use it (with $k \geq 3$) in training all RNNs.

## 3 ESTIMATING THE SCORE FUNCTION OF NOISY RNN ACTIVITY

Noisy RNNs trained to path-integrate implicitly learn the score function of their activity. Previous works have not examined the score function; in fact, they assume the distribution of RNN activity $p(\boldsymbol{r}(t))$ is stationary (Theorem 2), and thus the score function $\nabla_{\boldsymbol{r}(t)} \log p(\boldsymbol{r}(t))$ depends only on $\boldsymbol{r}(t)$ (Krishna et al., 2024). However, we refute this assumption, and in doing so reveal the role of leakage in path-integration: even if the RNN path-integrates a simple Gaussian process, the score function requires information beyond $\boldsymbol{r}(t)$, which it employs through linear leakage (decay) of $\boldsymbol{r}(t)$. This linearity suggests that leakage is useful for path-integration, which we confirm experimentally.

### 3.1 CHALLENGES IN SIMPLE DISTRIBUTIONS

First, we examine how the score function of optimal path-integrating RNN activity $\boldsymbol{r}(t)$ has nonstationarities that are challenging to perfectly estimate, even for simple Gaussian processes.

**Assumption 4.** *The observed states $\boldsymbol{s}(t)$ form some Gaussian process: $p(\boldsymbol{s}(t)) \sim \mathcal{N}\left(\boldsymbol{\mu}_{\boldsymbol{s}(t)}, \boldsymbol{\Sigma}_{\boldsymbol{s}(t)}\right)$.*

**Theorem 3.** *With Assumption 4, the score function of trained activity $\boldsymbol{r}(t)$ has a closed form which, while linear in $\boldsymbol{r}(t)$, is nonlinear with respect to the parameters of $p(\boldsymbol{s}(t))$ (see Appendix C.2):*

$$\sigma_r^2 \Delta t \, \nabla_{\boldsymbol{r}(t)} \log p(\boldsymbol{r}(t)) = -\underbrace{\sigma_r^2 \Delta t \left(\boldsymbol{I}\sigma_r^2 \Delta t + \boldsymbol{D}^{\dagger}\boldsymbol{\Sigma}_{\boldsymbol{s}(t)}(\boldsymbol{D}^{\dagger})^T\right)^{-1}}_{\boldsymbol{\Lambda}(t)} \left(\boldsymbol{r}(t) - \boldsymbol{D}^{\dagger}\boldsymbol{\mu}_{\boldsymbol{s}(t)}\right) \tag{15}$$

With $\boldsymbol{\Lambda}(t)$ as the *leakage matrix* of $\boldsymbol{r}(t)$, we can already gain some insight from Equation 15.

**Remark 1.** If $p(\boldsymbol{s}(t))$ is Gaussian, then the score of $p(\boldsymbol{r}(t))$ is simply a linear function of $\boldsymbol{r}(t)$, but its parameters are nonlinear functions of time, and are only as stationary as $p(\boldsymbol{s}(t))$.

**Remark 2.** The eigenvalues of the leakage matrix $\boldsymbol{\Lambda}(t)$ are always between 0 and 1 (see Appendix C.2). Moreover, the eigenvalues of $\boldsymbol{\Lambda}(t)$ and $\boldsymbol{D}^{\dagger}\boldsymbol{\Sigma}_{\boldsymbol{s}(t)}(\boldsymbol{D}^{\dagger})^T$ inversely correlate: strong decay of $\boldsymbol{r}(t)$ implies weak noise $\boldsymbol{\Sigma}_{\boldsymbol{r}(t)}$, and weak decay of $\boldsymbol{r}(t)$ implies strong noise $\boldsymbol{\Sigma}_{\boldsymbol{r}(t)}$.

To further illustrate how estimating the score function of $\boldsymbol{r}(t)$ (Equation 15) can be challenging, we now examine a scalar RNN $r_{ou}(t)$ trained to path-integrate Ornstein-Uhlenbeck processes.

**Assumption 5.** *The observed states follow a scalar Ornstein-Uhlenbeck process parameterized by $\theta, \mu, \sigma_s$: $s'_{ou}(t) = \theta(\mu - s_{ou}(t)) + \sigma_s \eta(t)$, where $p(s(0)) \sim \mathcal{N}(0, \sigma_0^2)$. In other words, $p(s_{ou}(t))$ parameterizes a directed random walk from $s_{ou}(0)$ to $\mu$.*

**Remark 3.** The score function of the optimal $r_{ou}(t)$ under Assumption 5 follows from Equation (27) (see Appendix D):

$$\sigma_r^2 \Delta t \, \nabla_{r_{ou}(t)} \log p(r_{ou}(t)) = \frac{-\sigma_r^2 \Delta t \left(r_{ou}(t) - \mu\left(1 - e^{-\theta t}\right)\right)}{\sigma_r^2 \Delta t + \frac{\sigma_s^2}{2\theta}\left(1 - e^{-2\theta t}\right) + \sigma_0^2 e^{-\theta t}} \tag{16}$$

The score function, and thus the optimal quiescent activity, is evidently complex and nonstationary: $\lim_{t \to 0} \sigma_r^2 \Delta t \nabla_{r_{ou}(t)} \log p(r_{ou}(t)) = -\frac{\sigma_r^2 \Delta t}{\sigma_r^2 \Delta t + \sigma_0^2} r_{ou}(t)$, while $\lim_{t \to \infty} \sigma_r^2 \Delta t \nabla_{r_{ou}(t)} \log p(r_{ou}(t)) = -\frac{\sigma_r^2 \Delta t}{\sigma_r^2 \Delta t + \sigma_s^2/2\theta}(r_{ou}(t) - \mu)$. While one could force

stationarity by implicitly assuming the process starts at $s_{ou}(0) = \mu$, or assuming the steady-state dynamics ($t \to \infty$) are the most important, we argue that any such approach would miss a fundamentally relevant aspect of the Ornstein-Uhlenbeck process for navigation: *intention*. Unlike the Wiener process (Appendix D), the Ornstein-Uhlenbeck process can describe a random walk that *intentionally* navigates from $s_u(0)$ to $\mu$, rather than one that simply wanders around $\mu$. Thus, for navigation, the non-stationary, or "early", dynamics of the Ornstein-Uhlenbeck process are the most salient. Given the relevance of non-stationary dynamics for navigation, our analyses focus on the entire course of replay dynamics (rather than steady-state distributions), examining properties like speed and path diversity (exploration).

### 3.2 The Advantage of Leakage

Theorem 1 and Remark 1 suggest that linear leakage may be a useful inductive bias for RNNs learning to path-integrate: the score function for a Gaussian process is linear with respect to $\boldsymbol{r}(t)$, although the parameters of said linearity are nonlinear functions of time. We examine the utility of leakage by comparing two RNNs trained to path-integrate: $\boldsymbol{r}(t + \Delta t) = \kappa \boldsymbol{r}(t) + f_1(\boldsymbol{r}(t), \boldsymbol{u}(t), \sigma_r \boldsymbol{\eta}(t))$ (RNN with leakage $0 < \kappa < 1$) and $\boldsymbol{r}(t + \Delta t) = f_2(\boldsymbol{r}(t), \boldsymbol{u}(t), \sigma_r \boldsymbol{\eta}(t))$ (RNN without leakage), where $f_1, f_2$ are shallow ReLU layers trained separately (see Appendix A.5). The second is more reminiscent of traditional RNNs in machine learning (e.g., ReLU or gated RNNs), which do not typically employ leakage. In Figure 2, we show that leakage is useful for path-integration, especially with masked training ($k > 1$). Our results suggest that leakage is a useful component of path-integrating RNNs, although other architectures may be able to successfully learn without it (e.g., the layer-norm RNN of Levenstein et al. (2024)).

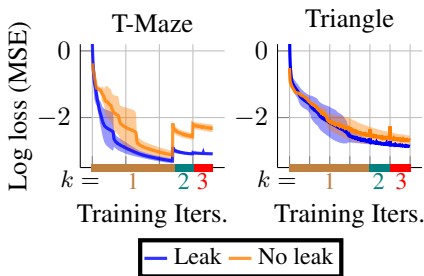

Figure 2: **Leakage helps path-integration.** Here we train RNNs on two tasks, ablating the leakage term. Leakage helps training, especially when losses increase with the masking difficulty $k$. Means are solid, standard deviations are faint.

## 4 Second-Order Langevin Sampling for Neural Replay

In the previous section, we examined the score function of trained path-integrating RNN activity. Now, we are assuming that trained RNNs have well-estimated the score function, and follow it during quiescent (internally-driven) activity to generate replay via Langevin dynamics. Here we ask how modulating the dynamics of such RNNs affects the distribution of replay—in other words, how changing the RNN dynamics biases the distribution of replay. We see that adaptation (negative feedback) incurs a non-ideal form of second-order Langevin sampling, so we propose a complementary alternative that explicitly performs underdamped second-order sampling.

### 4.1 Adaptation as Underdamped Langevin Dynamics

First we examine adaptation (negative feedback), as defined in Section 2.3.

**Proposition 1.** *Adding adaptation (Equation (13)) to a trained path-integrating RNN during quiescence (Equation (12)) incurs Langevin sampling with negative feedback:*

$$\Delta \boldsymbol{r}(t) = \sigma_r^2 \Delta t \, \nabla_{\boldsymbol{r}(t)} \log p(\boldsymbol{r}(t)) + \sqrt{2}\sigma_r \boldsymbol{\eta}(t) - \boldsymbol{c}(t), \quad \Delta \boldsymbol{c}(t) = \frac{1}{\tau_a}(-\boldsymbol{c}(t) + b_a \boldsymbol{r}(t)) \quad (17)$$

For the clearest illustration of the effects of adaptation, let us examine a stationary $p(\boldsymbol{r}(t))$—a simplification which we argued in Section 3.1 is not realistic, but is nonetheless intuitive.

**Assumption 6.** *The observed states are normal (Assumption 4) and stationary: $p(\boldsymbol{s}(t)) \sim \mathcal{N}(\boldsymbol{\mu}, \boldsymbol{\Sigma})$.*

**Theorem 4.** *Adding adaptation to an RNN trained to path-integrate states drawn from a stationary Gaussian distribution (Assumption 6 and Equation (17)) produces the following second-order*

*stochastic dynamics during quiescence (see Appendix E):*

$$\boldsymbol{r}''(t) = \left( \frac{b_a}{\tau_a} \boldsymbol{I} + \sigma_r^2 \Delta t \frac{d^2}{d\boldsymbol{r}(t)^2} \log p(\boldsymbol{r}(t)) \right) \boldsymbol{r}'(t) - \frac{b_a}{\tau_a} \sigma_r^2 \Delta t \, \nabla_{\boldsymbol{r}(t)} \log p(\boldsymbol{r}(t)) + \frac{1}{\tau_a} \boldsymbol{r}(t)$$
$$- \sigma \frac{b_a}{\tau_a} \boldsymbol{\eta}(t) + \sigma \boldsymbol{\eta}'(t) \tag{18}$$

Comparing Equation 18 with Equation 2, the two indeed resemble each other: adaptation seems to induce a form of underdamped Langevin dynamics. This may help explain the observed utility of adaptation for generating replay (Itskov et al., 2011; Levenstein et al., 2019; 2024). Moreover, since we established in Section 3 that the score function of even a basic stochastic process is difficult to estimate, the effectiveness of underdamped Langevin sampling for working with noisy gradients (Cheng et al., 2018) may be useful in realistic settings where $\nabla_{\boldsymbol{r}(t)} \log p(\boldsymbol{r}(t))$ is poorly estimated.

However, interpreting Equation (18) as underdamped Langevin sampling reveals some shortcomings thereof as a sampling method.

**Remark 4.** The coefficient of $\boldsymbol{r}'(t)$ is usually constant, and should be positive to ensure convergence (Pavliotis (2014), pg. 183), but $\frac{b_a}{\tau_a} \boldsymbol{I} + \nabla_{\boldsymbol{r}(t)} \log p(\boldsymbol{r}(t))$ is not constant and could be negative.

**Remark 5.** Underdamped Langevin sampling from $\boldsymbol{r}(t)$ should not have a negative sign in front of $\nabla_{\boldsymbol{r}(t)} \log p(\boldsymbol{r}(t))$ if the intention is to maximize $p(\boldsymbol{r}(t))$.

### 4.2 REPLAY VIA EXPLICIT UNDERDAMPED LANGEVIN DYNAMICS

While adaptation is a biologically plausible mechanism of performing a variant of underdamped Langevin dynamics in RNNs, we propose an alternative method, following Equation 3, to more clearly and directly perform underdamped Langevin sampling.

**Definition 3.** We implement *explicitly underdamped* dynamics via a momentum term $\boldsymbol{v}(t)$ controlled by friction $\lambda_v$ (when $\lambda_v = 1$, dynamics are identical to the overdamped case from Equation (4)):

$$\boldsymbol{v}(t + \Delta t) = (1 - \lambda_v)\boldsymbol{v}(t) + \underbrace{f(\boldsymbol{r}(t), \boldsymbol{u}(t), \sigma_r \boldsymbol{\eta}(t)) - \boldsymbol{r}(t)}_{\Delta \boldsymbol{r}(t) \text{ if } \lambda_v = 1}, \quad \Delta \boldsymbol{r}(t) = \boldsymbol{v}(t + \Delta t) \tag{19}$$

Our proposed sampling method is conceptually similar to RNNs with momentum (Nguyen et al., 2020): $\boldsymbol{v}(t)$ (i.e., the velocity of $\boldsymbol{r}(t)$) accumulates previous values when friction $\lambda_v \in [0, 1]$ is below 1[4]. Moreover, our sampling mechanism shares conceptual ties with synaptic facilitation, a phenomenon associated with temporal compression *in vivo* wherein inputs that successfully trigger output activity have a temporarily increased influence on output activity (Leibold et al., 2008; Thurley et al., 2008); we leave a possible implementation of underdampening via synaptic facilitation to future work. Lastly, our mechanism could relate to momentum observed in replay trajectories *in vivo* (Krause & Drugowitsch, 2022).

## 5 NUMERICAL RESULTS

Now we examine how adaptation and underdampening (Equations 13 and 19) bias replay distributions in trained path-integrating RNNs. We first see they counter each other: adaptation slows replay, while underdampening quickens it. Then we confirm that adaptation induces exploration, and show that underdampening does not prevent exploration, but rather complements it by increasing path lengths. We note here that replay trajectories are obtained by first randomly initializing the hidden state of the RNN, following which we collect activations for several unrolled timesteps in the absence of inputs but in the presence of noise. Further details are provided in Appendix A.1. Here we use ReLU and leaky ReLU activations in trained RNNs, as justified in Appendix A.2. In Appendix G we reproduce key findings with tanh RNNs.

**Experiments.** We have five tasks on which we train RNNs and then examine how incorporating sampling mechanisms post-training (adaptation strength $b_a$ and friction $\lambda_v$) affect replay (quiescent activity) statistics. For a detailed discussion of implementation, see Appendix A.

---

[4]For a comparison of $\lambda_v$ and the $\gamma$ term from Equations 3 and 2, see Appendix A.6.

1. 1D Ornstein-Uhlenbeck (OU) process (Figure 1): the optimal (with respect to Equation (8)) path-integrating RNN has a closed form (Equation (16)), which we use in lieu of a trained RNN.

2. 2D T-maze and triangle (Figure 3): we simulate each direction (two in T-maze, six in triangle) as a 2D OU process, and train a ReLU RNN to integrate 2D paths from every direction.

3. Rat trajectories (Appendix F, Figure 8): like Krishna et al. (2024), we simulate 512 place cells that encode 2D directed (biased) or undirected (unbiased) random walks from RatInABox (George et al., 2024), train a ReLU RNN to path-integrate place cell activity, and decode activity in 2D.

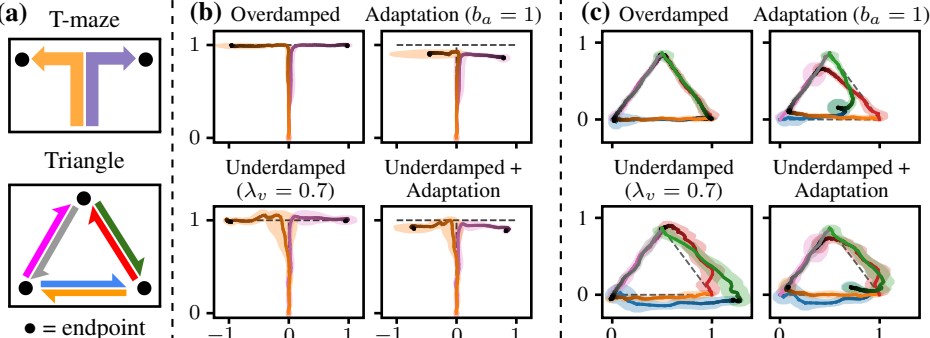

Figure 3: **Underdampening and adaptation counter each other.** Here we show replay from RNNs trained to path-integrate in T-maze or triangular environments. **(a)** Awake paths in each task form a mixture of Ornstein-Uhlenbeck processes, one for each direction of travel. Awake paths reach their endpoints and stay there. In **(b)** and **(c)** are mean replay paths (darkening over time) from T-maze and triangle tasks, simulated for the same time as awake paths. Standard deviations are faint, ideal path means are dashed. Like in Figure 1, adaptation slows convergence towards endpoints, underdampening quickens it. Also, the two mechanisms induce deviations that negate each other.

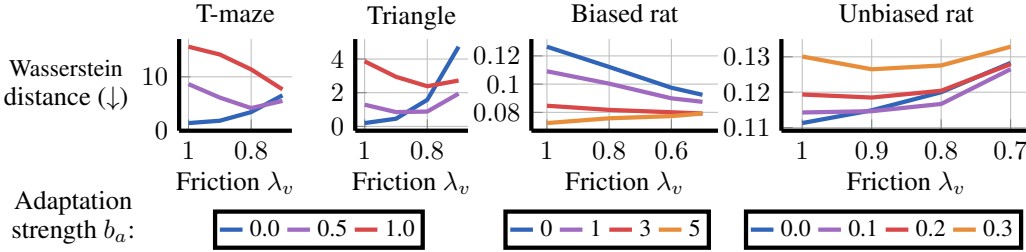

Figure 4: **Underdampening improves replay fidelity in the presence of adaptation.** We compute the Wasserstein distance (dissimilarity) between awake and replay path distributions ($p(\{\boldsymbol{s}(t)\}_{t=0}^{T})$ and $p(\{\boldsymbol{r}(t)\}_{t=0}^{T})$), varying friction and adaptation strength (see Appendix A for details). While the two mechanisms both generally increase this distance, underdampening ($\lambda_v < 1$) decreases it if adaptation is nonzero. Like in Figure 3, underdampening counters adaptation-induced deviations.

**Underdampening and adaptation seem to counter each other.** We initially observe that the two modifiers counteract each other: in Figures 1 and 3, adaptation repels trajectories away from attractors (such as endpoints), while underdampening accelerates trajectories towards them. This makes sense given Remark 5 and the nature of underdamped sampling. Then we measure how the two modifiers affect the similarity between replay (internally-driven) and awake (observed) trajectories. Figure 4 shows they both generally decrease the similarity to awake trajectories, but they do counter each other insofar as underdampening, in the presence of adaptation, increases similarity to awake trajectories. Thus far, underdampening counters adaptation qualitatively and statistically.

**Adaptation slows replay, underdampening accelerates it.** Next, we examine a key component of replay: speed. In Figures 4 and 5, we generally see that adaptation slows replay or increases the

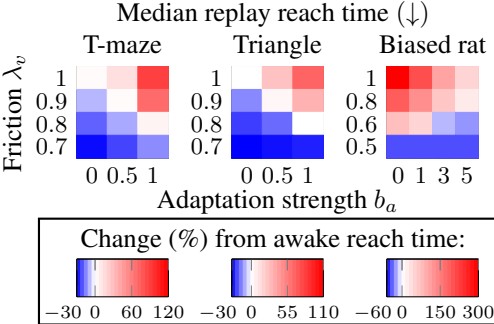

Figure 5: **Underdampening temporally compresses replay.** We calculate how long it takes awake and replay trajectories to reach their endpoints. Underdampening ($\lambda_v < 1$) not only shortens this reach time, but makes it smaller than that of awake paths, *temporally compressing* awake activity. See Appendix F Figure 9 for mean reach times. We do not include unbiased rat trajectories because they do not have defined endpoints, but we do confirm in Appendix F Figure 10 that underdampening quickens them.

dissimilarity between replay and awake trajectories. The exception is the biased rat trajectories, since they are the only task where all trajectories reach the same common goal. Such path distributions resemble Ornstein-Uhlenbeck processes with steady-state mean $\mu = 0$: in such cases, adaptation actually accelerates convergence towards the steady-state. In Figure 5, we see that underdampening increases replay speed, performing temporal compression relative to awake activity.

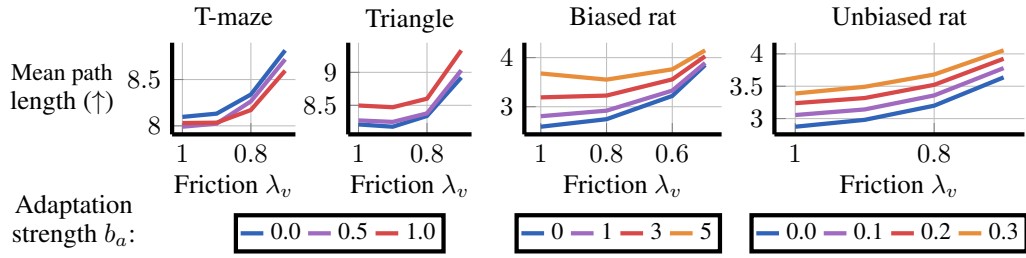

Figure 6: **On average, underdampening increases exploration via path length.** In here and Figure 7 we simulate replay paths for more time than awake paths, varying friction and adaptation strength. Shown above, underdampening ($\lambda_v < 1$) increases the average length of replay paths. It may do so via increased replay speeds (Appendix F Figure 10) or via additional path transitions (Figure 7); the latter is how adaptation increases path length.

**Underdampening complements exploration from adaptation.** We have established that underdampening counters adaptation, both qualitatively and in terms of speed. However, we do not want to merely propose a mechanism that undoes adaptation. Adaptation can induce exploration, i.e., prevent replay trajectories from getting stuck in attractors. We find that underdampening does not on average prevent adaptation-induced exploration. In fact, underdampening complements adaptation for exploration: underdamped paths travel farther (Figures 6 and 7), have more variance (Appendix F Figure 11), and generally exhibit the same, if not more, exploratory behavior as they did with only adaptation (Figure 7). Underdampening maintains exploration while counteracting the slowness from adaptation.

## 6 CONCLUSIONS AND FUTURE WORK

We have re-applied Langevin sampling theory to replay in sequential predictive learning networks, producing theoretically and confirming empirically three key insights: (1) estimating the per-timestep score function of RNN activity is challenging, but does benefit from linear leakage; (2) adaptation (negative feedback) is a variant of underdamped Langevin sampling that encourages exploration (as shown in prior works) but also slows replay; (3) our new underdampening mechanism (momentum) temporally compresses replay while also increasing (on average) exploration from adaptation. These findings improve our understanding of biological neural networks that produce replay, like the hippocampus. Our proposed underdampening mechanism via momentum could be tied to short-term facilitation (which can be probed experimentally), connected to specific subregions in the hippocampus (like Chen et al. (2024)), or refined through insights from existing RNNs with momentum (Nguyen et al., 2020). Future efforts could confirm our findings in more complex environments Wood

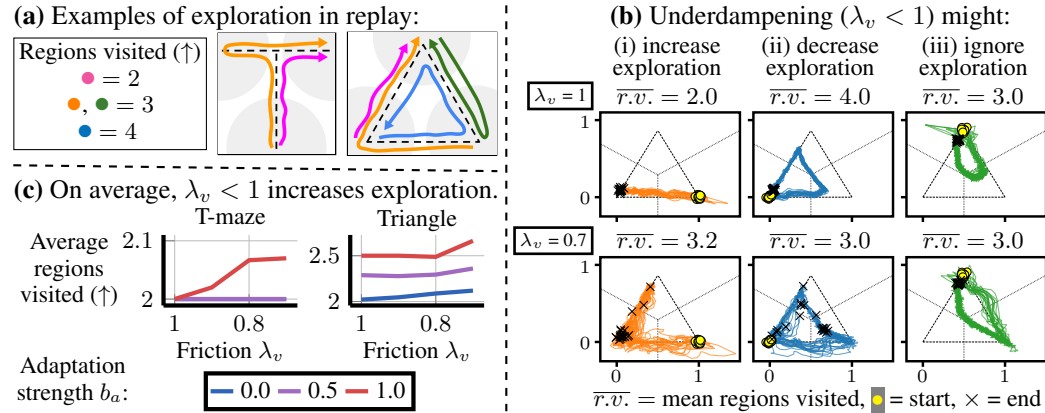

Figure 7: **On average, underdampening increases exploration via regions visited.** Another relevant aspect of exploration is transitions not present in awake activity. **(a)** Here we quantify these transitions via *regions visited*. Each region is an area where every point is closest to the same endpoint. In our T-maze and triangle tasks, awake paths go from starting points to endpoints and stay there, visiting two regions total (starting and ending). Meanwhile, adaptation ($b_a = 1$) can make replay paths visit multiple endpoints (regions visited > 2). **(b)** When we use adaptation and underdampening ($\lambda_v < 1$), we see that replay paths might visit (i) more, (ii) fewer, or (iii) the same regions (see Appendix F Figure 12 for more details). **(c)** However, on average, across several different trained models, underdampening increases regions visited, thus increasing adaptation.

et al. (2018); Levenstein et al. (2024). Like preceding works, our network models are rate-based, but extending our work to spike-based models of replay and sequential predictive learning (Saponati & Vinck, 2023; Asabuki & Fukai, 2025; Bono et al., 2023) would be interesting (this may involve extending Langevin sampling to inhomogeneous Poisson processes). There still remain several avenues for connecting Langevin sampling to neural replay.

## REPRODUCIBILITY STATEMENT

For our theoretical contributions, i.e., Theorems 3 and 4, we provide proofs in Appendix C and E (proofs for Theorems 1 and 2 may be found in Section 2 of Krishna et al. (2024)). Details on our numerical experiments, including associated hyperparameters and implementation details are provided in Appendix A. Our code has been submitted as Supplementary Material (see README.md for instructions) and will be made publicly available upon publication.

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

SUPPLEMENTARY MATERIAL

# A  METHODS

## A.1  EXPERIMENTS

### A.1.1  ORNSTEIN-UHLENBECK

Awake trajectories $s(t)$ are simulated with $\Delta t = 0.02, \sigma_s = 0.1, \sigma_0 = 0.2, \theta = 2, \mu = 5$ for $T = 100$ iterations. We used Equation (16) as the deterministic component of $\Delta r(t)$ since a 1D Orstein-Uhlenbeck process admits a closed-form expression for $\frac{d}{dr(t)} \log p(r(t))$. In Figure 1, underdampening corresponds to $\lambda_v = 0.5$, while adaptation corresponds to $b_a = 1, \tau_a = 100$.

### A.1.2  T-MAZE AND TRIANGLE

**Task description.** In the 2D T-maze and triangle tasks, we simulate directed random walks (Ornstein-Uhlenbeck processes) along several directions, and train the RNN to path-integrate these walks from their velocities. In the T-maze task, there are two directions of travel: both start from the origin and go up, then one goes left and the other goes right (orange and purple, respectively, in Figure 3). Meanwhile, in the triangle task there are six directions: denoting the three corners of the equilateral triangle as $A = (0,0), B = (1,0)$, and $C = (1/2, \sqrt{3}/2)$, respectively, these directions are $\overrightarrow{AB}, \overrightarrow{BC}, \overrightarrow{CA}, \overrightarrow{AC}, \overrightarrow{CB}, \overrightarrow{BA}$, shown in Figure 3 as blue, red, gray, pink, green, and orange, respectively. Awake and replay paths last 100 timesteps, unless exploration is being measured, in which replay paths are allowed to last 400 timesteps.

**Architecture and training.** For both tasks, we used shallow leaky-ReLU RNNs with linear output projections, as described in Appendix A.5. In the T-maze task, we used 20 hidden neurons, while in the triangle task, we used 40. We use masking difficulty $k = 3$ in a progressive curriculum. Triangle-task RNNs are trained with $k = 1$ for $20,000$ epochs, then at $k = 2$ and $k = 3$ for $5,000$ epochs each; T-maze-task RNNs are trained likewise, but with fewer epochs ($12,000$ at $k = 1$ and $5,000$ at $k = 2$ and $k = 3$ each). In each task, we train 5 RNNs with different seeds.

**Hidden state initialization.** In each task, multiple directions start from the same point in space (for example, $\overrightarrow{AB}$ and $\overrightarrow{AC}$ in the triangle task), so we add onto initial hidden states some random vectors, orthogonal to the 2D output projection, specific to each direction. For example, hidden states for $\overrightarrow{AB}$ and $\overrightarrow{AC}$ paths are both initialized to start near $A$, but $\overrightarrow{AB}$ paths also start with a fixed random vector $\boldsymbol{\eta}_{AB}$ added to their initial hidden states, whereas $\overrightarrow{AC}$ have a different fixed random vector $\boldsymbol{\eta}_{AC}$ added to their initial hidden states. This notion of initializing hidden states in directions orthogonal to output projections has been previously discussed in computational neuroscientific contexts (Churchland & Shenoy, 2024), and recent evidence suggests that memories (which replay is similar to) have uniquely identifiable, output-orthogonal patterns of neural activation (Chettih et al., 2024).

### A.1.3  RAT PLACE CELL TRAJECTORIES

Our unbiased (undirected random walks within a 2D box) and biased (directed random walks that head towards the center of a 2D box) rat place cell trajectory experiments are identical to those of Krishna et al. (2024), with 5 different ReLU RNN seeds per task, but with two modifications:

1. We add masked training, with difficulties $k = 3$ in the biased task and $k = 6$ in the unbiased task. We use a higher masking difficulty in the unbiased task to encourage longer replay trajectories; at lower values of $k$, unbiased replay paths are much shorter than awake trajectories. In Langevin sampling terms, we conjecture this is because the score function $\nabla_{\boldsymbol{r}(t)} \log p(\boldsymbol{r}(t))$ is fairly weak since unbiased trajectories are uniformly distributed[5].

---

[5]This relative weakness the unbiased activity score function (especially compared to the biased activity score function, where trajectories all go towards the center of a 2D box) is also why we use relatively small adaptation strengths $b_a$ in simulating replay from unbiased-task RNNs.

2. We use a slower, but more detailed, decoder. Krishna et al. (2024) decode position at a given timestep as the average position of the top 3 most active place cells, which is fast but effectively quantizes decoded positions. We use this method to initialize our positions, but then we optimize our positions via gradient descent to minimize the mean-squared error between observed place cell activity and the place cell activity that would correspond to these optimized positions (this is made possible by our knowledge of the exact place cell activity function). Our procedure takes much longer, but results in truly continuous replay trajectories, avoiding any significant quantization.

## A.2 HYPERPARAMETERS

### A.2.1 ACTIVATION FUNCTION

In our work, we have mostly used RNNs with ReLU or leaky ReLU activation functions (except in Appendix G); our reasons for doing so are threefold. The first is that ReLU or ReLU-like activations, unlike tanh activations, do not always saturate, which can mitigate vanishing gradients in RNNs (although care must be taken to avoid exploding gradients). The second is that, compared to RNNs with multiplicative gating interactions (e.g., GRUs and LSTMs), ReLU RNNs are more biologically plausible insofar as they can be easily interpreted as the nonnegative, sparse firing rates of spiking networks. The third is that most other previous works in sequential predictive coding tend to use ReLU activations, and have shown no significant differences in results when using more complex nonlinearities:

- Ali et al. (2022), some of the first authors to report the emergence of predictive representations emerging in RNNs, use ReLU activations.
- Krishna et al. (2024) primarily use ReLU networks, and report similar results with GRU networks.
- Levenstein et al. (2024) use ReLU activations in conjunction with layer normalization, but report similar results when layer normalization is removed.
- Chen et al. (2024) use tanh for must results, but when they care about the emergence of sparse, localized, nonnegative activations (i.e., place cells), they use ReLU activations.
- Sorscher et al. (2019); Xu et al. (2025), and Zhang et al. (2022) were all interested in the emergence of grid cells in predictive representations and primarily used ReLU or ReLU-like activations; those that tried using other nonlinearities did not observe notable differences in results.
- Tang et al. (2024), who were also interested in grid cells, use both ReLU and tanh activations, and also report no notable differences between the two.
- Tang et al. (2023) use linear networks, and notice no significant difference when using tanh activations.

## A.3 METRICS

### A.3.1 WASSERSTEIN DISTANCE

We seek to compare distributions of trajectories. In most of our tasks, these are objects of dimension $T \times 2$, where $T$ is the number of timesteps (recall that T-maze and triangle tasks are already in 2D, while rat trajectories are analyzed after being projected into 2D space). To compare such high-dimensional distributions, we use Wasserstein distances (Panaretos & Zemel, 2019):

- In the T-maze and triangle tasks, we first calculate the Wasserstein distance between awake and replay paths belonging to the same direction (e.g., $\overrightarrow{AB}$ in the triangle task). Since paths along the same direction should resemble, if not obey, Gaussian processes, these paths should approximately be normally distributed, and so we can use the closed-form equation for the Wasserstein distance between two normal distributions. After computing Wasserstein distances for each direction, we take the average as our final distance. KL divergence might have also worked in this task, but in practice the covariance matrices of the $(T \times 2)$-dimensional distributions were singular.
- In the rat experiments, decoded 2D trajectories are not easily decomposed into groups of Gaussian processes, so instead we apply sliced Wasserstein distances (Bonneel et al., 2015) to compare the $(T \times 2)$-dimensional distributions. Sliced Wasserstein distances are essentially calculated by

taking many random projections of two distributions onto 1D, where Wasserstein distances have a closed form. Computing KL divergences instead would have been challenging, since rat path distributions are high-dimensional and do not admit straightforward estimations of probability density.

### A.3.2 REACH TIMES

In the T-maze, triangle, and biased rat tasks, awake trajectories have clearly defined endpoints that they reach. Replay trajectories also aim for these endpoints, although second-order dynamics modifiers like adaptation and underdampening affect how quickly or how closely replay paths reach them. We quantify these changes by measuring the timesteps required for replay paths to get within 10% of their endpoints (for example, the timesteps required for $\overrightarrow{AB}$ paths to get within a $0.1|\overrightarrow{AB}|$ of $B$).

### A.3.3 PATH LENGTHS

One way we quantify exploration in replay paths is through path length. We calculate this simply as the sum of velocity magnitudes.

### A.4 REGIONS VISITED

Another way we quantify exploration in the T-maze and traingle tasks is through *regions visited*. As explained in Figure 7, *regions* are portions of input space where all points are closest to the same endpoint. In the triangle task, there are 3 endpoints (which all also act as initial points), while in the T-maze there are 2 endpoints and 1 shared starting point, which for the purposes of region assignment we also consider an "endpoint". Thus, in each task there are 3 endpoints, and thus 3 regions. Awake trajectories go from one region to another and stay there, but replay trajectories might proceed to visit more regions. We define a "visit" as a contiguous presence within a single region for at least 10 timesteps.

### A.5 DISCRETIZATION AND IMPLEMENTATION OF NOISY RNNS

While biological neural networks are continuous-time systems, RNNs are in practice implemented discretely. In order to incorporate second-order processes like adaptation and underdamped sampling in RNNs, we must discretize them, which we do as follows:

$$\Delta\widetilde{r}(t) = f(r(t), u(t), \sigma_r\eta(t)) - r(t) \tag{20}$$

$$c(t + \Delta t) = c(t) + \frac{1}{\tau_a}(-c(t) + b_a r(t)) \tag{21}$$

$$v(t + \Delta t) = (1 - \lambda_v)v(t) + \Delta\widetilde{r}(t) \tag{22}$$

$$r(t + \Delta t) = r(t) - c(t) + v(t + \Delta t) \tag{23}$$

We train and sample from ReLU RNNs: $f(r(t), u(t), \sigma_r\eta(t)) = \kappa r(t) + \text{ReLU}(W_r r(t) + W_{in}u(t) + \sigma_r\eta(t))$ (where $k \in [0, 1]$ is also learnable) unless otherwise stated (a choice justified in Appendix A.2). In the absence of directed inputs $u(t)$, $f(r(t), u(t), \sigma_r\eta(t))$ should be roughly equivalent to the noisy score function from Equation 12.

If the adaptation strength $b_a = 0$, then there is no adaptation ($c(t) = 0$). As for our *friction* term $\lambda_v \in [0, 1]$, if $\lambda_v = 1$, then there is no underdamped sampling since $v(t) = \Delta r(t) + \sigma_r\eta(t)$ no longer accumulates previous values of $v(t)$. In other words, the friction term $\lambda_v$ allows for smooth interpolation between overdamped and underdamped sampling.

Note that, unlike $c(t + \Delta t)$ and $v(t + \Delta t)$, $r(t + \Delta t)$ depends on another term calculated at $t + \Delta t$. This is a symplectic Euler discretization of the second-order dynamics, which we employ for to ensure the stability of interactions between $r(t)$ and its momentum $v(t)$. The negative feedback $c(t)$ is much more stable, so its discretization goes unchanged.

Across all trained RNN experiments we use $\tau_a = 0, \lambda_v = 1$ for awake activity (i.e., training). When generating replay, we always use $\tau_a = 100$, and either $T = 100, 400$, or $500$ timesteps, depending

on whether we are measuring fidelity and speed ($T = 100$, like in Figures 3, 4, 5) or exploration ($T = 400$ for T-maze and triangle, $T = 500$ for rat tasks, like in Figures 6, 7).

### A.6 DEVIATIONS FROM TRADITIONAL LANGEVIN SAMPLING

We must note a few minor distinctions between our replay RNNs and traditional Langevin sampling.

**Stationarity.** We treat neural replay as a sequence of events, and are thus interested in the joint distribution of replay activity at all timesteps $p\left(\{r(t)\}_{t=1}^{T}\right)$. This distribution, however, is not necessarily stationary across $t$. Stationarity would imply that $p(r(t))$ has no need or intention of traversing along any meaningful path. While a path-integrating RNN does perform gradient ascent along $\log p(r(t))$, it does so in a piecewise manner along $t$ rather than jointly along all $t$ simultaneously. This variation on gradient ascent, in combination with the non-stationarity of $p(r(t))$, means that Langevin sampling guarantees do not hold. For this reason, we make modifications to our RNNs that under stationary Langevin sampling theory might seem unprincipled.

**Underdampening.** The friction term $\lambda_v$ applied to the velocity $v(t)$ is subtly different from $\gamma$ in Equations 3 and 2. $\lambda_v \in [0, 1]$ can be interpreted as trying to map $\gamma \in [0, \infty)$ onto $[0, 1]$. Like $\gamma \to \infty$, $\lambda_v \to 1$ removes any dependency of $v(t + \Delta t)$ on $v(t)$.

**Noise scaling.** RNN replay, unlike Langevin sampling, is sensitive to the variance of noise used. We found that omitting the $\sqrt{2}$ factor in front of $\sigma_r \eta(t)$ worked best. For this reason, we also omitted $\lambda_v$ as a scaling term on $\sigma_r \eta(t)$. If $\lambda_v$ is allowed to modulate $\sqrt{2}\sigma_r \eta(t)$, we found that it is easy to show improvements in replay fidelity when $\lambda_v < 1$, but we found these to come from noise scaling rather than from underdampening or momentum as mechanisms.

**Relation to diffusion models.** Throughout this work we treat RNNs as generative models using a variant of Langevin sampling. Expressive generative models, in particular *diffusion models* whose activity resembles Langevin sampling, have received much attention from the machine learning community (Alemohammad et al., 2024; Luzi et al., 2024; Yang et al., 2023; Croitoru et al., 2023; Li et al., 2022; Gozalo-Brizuela & Garrido-Merchan, 2023; Song & Ermon, 2019). The fundamental differences between our RNNs and diffusion models are twofold. The first is that each timestep of a generated replay trajectory is generated sequentially and with only one effective step along the gradient of log-likelihood, whereas a diffusion model would generate all timesteps of a path simultaneously and with many steps along the gradient of log-likelihood. The second is that diffusion models use time-varying noise levels (annealed dynamics) and are explicitly conditioned on time as an input, whereas our RNNs use the same noise level across time and are never explicitly conditioned on time.

## B OVERVIEW OF BACKGROUND

Throughout the text we make use of notions including sampling, Langevin dynamics, replay, path-integration, awake and quiescent activity, adaptation, and masked training. Here we gather these notions, which have been defined in various previous work, as follows:

1. Our work examines how trained noisy RNNs can act as generative models in the absence of inputs, which prior work including Krishna et al. (2024) consider to be models of replay in neuronal circuits. Accordingly, we define Langevin dynamics, with (Equation (1)) and without (Equations 2 and 3) momentum, as an iterative process by which an agent could attempt to sample from an unknown distribution $p(x)$ using only knowledge of its *score function* $\nabla \log p(x)$, which can be learned from noisy observations drawn from the distribution. The following points provide more detail on the sampling process and why Langevin dynamics arise:

   - Biological networks such as those underlying navigation must accurately estimate environmental state variables such as position from observations of self-motion, even in the presence of intrinsic noise.
   - This noise induces a distribution over network states, i.e., activity, however, it is unknown exactly what the true noise distribution is.

- Accurate state estimation would thus require the network to optimally remove intrinsic noise without access to the exact parameters of the noise distribution, but given noisy network states.

- This optimal solution can be accomplished by learning the score function of noisy network states and using it to denoise network states over time as additional inputs and noise are presented (Miyasawa, 1961).

- When the network receives no inputs, its noise-driven dynamics still attempt to remove intrinsic noise, but the learned score function is for the distribution of waking task-like activity. This causes the network states to resemble actual samples drawn from the distribution over network states in the presence of inputs. The theoretical results show that the network dynamics during this quiescent state carry out sampling by Langevin dynamics, i.e., using the score function of the network's activity distribution during task performance to yield quiescent network activity that resembles waking activity.

2. Biological neural circuits like the hippocampus can navigate through environments when awake, and recall navigatory episodes during rest. Accordingly, we summarize the proof from Krishna et al. (2024) that a noisy RNN trained on a biologically relevant navigation task like path-integration can act as a generative model.

   - Definitions 1 and 2 simply express that our RNNs have nonlinear dynamics and internal noise, and are trained to estimate a signal $s(t)$ by integrating observations of $s'(t)$. This phenomenon, known as path-integration, underlies navigation and is used to estimate one's position from self-motion cues.

   - Assumption 1 asserts that RNN dynamics can be decomposed into additive components. Assumption 2 simply states that the optimal RNN dynamics would lead to accurate path-integration, such that the conditional probability distribution over network states (activity) given environmental states (position) would be Gaussian and identical to the additive noise in our networks. That is, the variability over network states would only be due to intrinsic noise in the system and have the same statistics. Finally, Assumption 3 assumes that the RNN dynamics are greedily optimal for path-integration. Greedy optimization is a sensible way of partitioning effort across time in path-integration: the network does the best that it can at each timestep, assuming that at each previous timestep the best possible job has been done.

   - Theorem 1 states that path-integrative RNN dynamics will use the score function of waking activity to remove intrinsic noise from the system and use inputs to update state estimates. This is a two-step greedily optimal solution for path integration in the presence of intrinsic noise: first, intrinsic noise must be removed, following which the inputs must be used to update the estimate of the current environmental state. Theorem 2 shows that these dynamics will result in Langevin sampling from the distribution of waking activity in the absence of any inputs, i.e., analogous to sleep. This means that the quiescent network dynamics sample neural states from the same distribution as waking, task-like activity, and can thus represent sequences like those during awake task performance, i.e., leading to replay.

3. Finally, we define existing methods of modulating RNN activity or training that affect RNN replay distributions: adaptation (negative feedback), which encourages diversity in replay paths, and masked training, which encourages coherence in replay.

## C    SCORE FUNCTIONS OF GAUSSIAN DISTRIBUTIONS

For any matrix calculus involved, we use denominator layout.

### C.1 MULTIVARIATE GAUSSIAN DISTRIBUTION

Let's suppose $\boldsymbol{r} \sim \mathcal{N}(\boldsymbol{\mu}, \boldsymbol{\Sigma})$. If $\boldsymbol{r} \in \mathbb{R}^d$, then:

$$p(\boldsymbol{r}) = \frac{1}{\sqrt{(2\pi)^d |\boldsymbol{\Sigma}|}} \exp\left(-\frac{1}{2}(\boldsymbol{r}-\boldsymbol{\mu})^T \boldsymbol{\Sigma}^{-1}(\boldsymbol{r}-\boldsymbol{\mu})\right) \tag{24}$$

$$\log p(\boldsymbol{r}) \propto -\frac{1}{2}(\boldsymbol{r}-\boldsymbol{\mu})^T \boldsymbol{\Sigma}^{-1}(\boldsymbol{r}-\boldsymbol{\mu}) \tag{25}$$

$$\nabla_{\boldsymbol{r}} \log p(\boldsymbol{r}) = -\frac{1}{2}((\boldsymbol{\Sigma}^{-1})^T + \boldsymbol{\Sigma}^{-1})(\boldsymbol{r}-\boldsymbol{\mu}) \tag{26}$$

$$= -\boldsymbol{\Sigma}^{-1}(\boldsymbol{r}-\boldsymbol{\mu}) \tag{27}$$

$$= -\sigma^{-2}(r-\mu) \text{ if } r \in \mathbb{R} \tag{28}$$

### C.2 SCORE FUNCTION OF $r(t)$ FOR GAUSSIAN $s(t)$

Recall that $p(\boldsymbol{r}(t)|\boldsymbol{s}(t)) \sim \mathcal{N}(\boldsymbol{D}^\dagger \boldsymbol{s}(t), \boldsymbol{I}\sigma_r^2 \Delta t)$ from Equation 7. If we suppose that $\boldsymbol{s}(t)$ is normally distributed with mean and covariance $\boldsymbol{\mu}_{\boldsymbol{s}(t)}, \boldsymbol{\Sigma}_{\boldsymbol{s}(t)}$, then we can obtain $p(\boldsymbol{r}(t))$:

$$p(\boldsymbol{r}(t)) \sim \mathcal{N}(\boldsymbol{D}^\dagger \boldsymbol{\mu}_{\boldsymbol{s}(t)}, \boldsymbol{I}\sigma_r^2 \Delta t + \boldsymbol{D}^\dagger \boldsymbol{\Sigma}_{\boldsymbol{s}(t)} (\boldsymbol{D}^\dagger)^T), \tag{29}$$

which we can plug into Equation 27 to get $\nabla_{\boldsymbol{r}(t)} \log p(\boldsymbol{r}(t))$:

$$\nabla_{\boldsymbol{r}(t)} \log p(\boldsymbol{r}(t)) = -\left(\boldsymbol{I}\sigma_r^2 \Delta t + \boldsymbol{D}^\dagger \boldsymbol{\Sigma}_{\boldsymbol{s}(t)}(\boldsymbol{D}^\dagger)^T\right)^{-1}\left(\boldsymbol{r}(t) - \boldsymbol{D}^\dagger \boldsymbol{\mu}_{\boldsymbol{s}(t)}\right) \tag{30}$$

Moreover, we can use the above score function to calculate the optimal $\Delta \boldsymbol{r}(t+\Delta t)$ in Equation 11:

$$\Delta \boldsymbol{r}^*(t+\Delta t) = \sigma_r^2 \Delta t \left(\boldsymbol{I}\sigma_r^2 \Delta t + \boldsymbol{D}^\dagger \boldsymbol{\Sigma}_{\boldsymbol{s}(t)}(\boldsymbol{D}^\dagger)^T\right)^{-1}\left(-\boldsymbol{r}(t) + \boldsymbol{D}^\dagger \boldsymbol{\mu}_{\boldsymbol{s}(t)}\right)$$
$$+ \boldsymbol{D}^\dagger \boldsymbol{s}'(t)\Delta t + \sigma_r \boldsymbol{\eta}(t) \tag{31}$$

Some properties of the leakage matrix $\sigma_r^2 \Delta t \left(\boldsymbol{I}\sigma_r^2 \Delta t + \boldsymbol{D}^\dagger \boldsymbol{\Sigma}_{\boldsymbol{s}(t)}(\boldsymbol{D}^\dagger)^T\right)$ include:

1. The covariance matrix $\boldsymbol{\Sigma}_{\boldsymbol{s}(t)}$ is positive semidefinite (PSD): all its eigenvalues are $\geq 0$.

2. $\boldsymbol{D}^\dagger \boldsymbol{\Sigma}_{\boldsymbol{s}(t)}(\boldsymbol{D}^\dagger)^T$ is also PSD [6], symmetric, and therefore diagonalizable.

3. The eigenvalues of $(\boldsymbol{I}\sigma_r^2 \Delta t + \boldsymbol{D}^\dagger \boldsymbol{\Sigma}_{\boldsymbol{s}(t)}(\boldsymbol{D}^\dagger)^T)^{-1}$ are thus all $\leq (\sigma_r^2 \Delta t)^{-1}$ [7].

4. The eigenvalues of $\sigma_r^2 \Delta t(\boldsymbol{I}\sigma_r^2 \Delta t + \boldsymbol{D}^\dagger \boldsymbol{\Sigma}_{\boldsymbol{s}(t)}(\boldsymbol{D}^\dagger)^T)^{-1}$ are thus all $\leq 1$.

5. If the off-diagonal entries of the leakage matrix above are sufficiently small in magnitude, then all the diagonal entries should be less than 1, as justified by the Gershgorin Circle Theorem. In fact, if the leakage matrix is diagonal, then it must have all values less than 1 (which could be achieved via sigmoid functions or perhaps spectral normalization).

6. As for interpretation, smaller leakage eigenvalues means higher eigenvalues of $\boldsymbol{D}^\dagger \boldsymbol{\Sigma}_{\boldsymbol{s}(t)}(\boldsymbol{D}^\dagger)^T$, or essentially, more noise. The maximum determinant of the leakage matrix is 1, when there is essentially no noise in $\boldsymbol{s}(t)$.

## D ADDITIONAL SCORE FUNCTION RESULTS

**Wiener Processes.** One simple stochastic process is the Wiener process, which in terms of navigation represents an undirected random walk ($\theta = 0$). Assuming $s_w(0) = 0$, then $s_w(t) \sim \mathcal{N}(0, \sigma_s^2 t)$, and therefore $p(r_w(t)) \sim \mathcal{N}(0, \sigma_s^2 t + \sigma_r^2 \Delta t)$ from Equation 7, producing the following score:

$$\sigma_r^2 \Delta t \, \nabla_{r_w(t)} \log p(r_w(t)) = \sigma_r^2 \Delta t \frac{-r_w(t)}{\sigma_s^2 t + \sigma_r^2 \Delta t} \tag{32}$$

Even from a simple Wiener process, we observe that the per-timestep optimal score is not constant with respect to $t$: at $t = 0$, it equals $-r_w(t)$, while as $t$ approaches $\infty$, it approaches 0.

---

[6]Proof: If $\boldsymbol{B}$ is PSD, then $x^T \boldsymbol{A}\boldsymbol{B}\boldsymbol{A}^T x = (\boldsymbol{A}^T x)^T \boldsymbol{B}(\boldsymbol{A}^T x) = v^T \boldsymbol{B}v \geq 0$.

[7]Proof: For diagonalizable $\boldsymbol{A}$, the $i$-th eigenvalue of $(\lambda \boldsymbol{I} + \boldsymbol{A})^{-1} = (\boldsymbol{Q}(\lambda \boldsymbol{I} + \boldsymbol{\Lambda})\boldsymbol{Q}^{-1})^{-1}$ is equal to $(\lambda + \boldsymbol{\Lambda}_{ii})^{-1}$, which can be no larger than $\lambda^{-1}$ if $\boldsymbol{A}$ is PSD.

**Ornstein-Uhlenbeck Processes.** Now we incorporate non-zero leakage ($\theta > 0$) to describe a directed random walk navigating from an arbitrary starting point $s_{ou}(0)$ towards a mean destination $\mu$. If $p(s_{ou}(0)) \sim \mathcal{N}(0, \sigma_0^2)$, then $p(s_{ou}(t)) \sim \mathcal{N}\left(\mu(1 - e^{-\theta t}), \frac{\sigma_s^2}{2\theta}(1 - e^{-2\theta t}) + \sigma_0^2 e^{-\theta t}\right)$, and the score function is:

$$\sigma_r^2 \Delta t \, \nabla_{r_{ou}(t)} \log p(r_{ou}(t)) = \sigma_r^2 \Delta t \frac{-(r_{ou}(t) - \mu(1 - e^{-\theta t}))}{\frac{\sigma_s^2}{2\theta}(1 - e^{-2\theta t}) + \sigma_0^2 e^{-\theta t} + \sigma_r^2 \Delta t} \tag{33}$$

# E   ADAPTATION AS A SECOND-ORDER STOCHASTIC DIFFERENTIAL EQUATION

Let us first combine the following two coupled linear stochastic differential equations into one second-order equation:

$$dX_t = (AX_t + BY_t + M)dt + \sigma dB_t, \quad dY_t = (CX_t + DY_t)dt \tag{34}$$

$$d^2 X_t = A dX_t + B dY_t + \sigma d(dB_t) \tag{35}$$

$$= A dX_t + B dY_t + \sigma d^2 B_t \tag{36}$$

$$= A dX_t + B(CX_t + DY_t)dt + \sigma d^2 B_t, \; Y_t = \frac{1}{Bdt}(dX_t - \sigma dB_t - AX_t dt - M dt) \tag{37}$$

$$= A dX_t + BCX_t dt + BD \frac{1}{Bdt}(dX_t - \sigma dB_t - AX_t dt - M dt)dt + \sigma d^2 B_t \tag{38}$$

$$= A dX_t + BCX_t dt + D(dX_t - \sigma dB_t - AX_t dt - M dt) + \sigma d^2 B_t \tag{39}$$

$$= (A + D)dX_t + (BC - AD)X_t dt - DM dt - \sigma D dB_t + \sigma d^2 B_t \tag{40}$$

Replacing all variables involved (except $dt, \sigma$) with matrices and vectors yields the same equation as long as $\boldsymbol{B}$ is invertible:

$$d^2 \boldsymbol{x}_t = (\boldsymbol{A} + \boldsymbol{D})d\boldsymbol{x}_t + (\boldsymbol{BC} - \boldsymbol{AD})\boldsymbol{x}_t dt - \boldsymbol{Dm}dt - \sigma \boldsymbol{D} d\boldsymbol{B}_t + \sigma d^2 \boldsymbol{B}_t \tag{41}$$

For consistency with the notation used throughout the paper, the equation above can be written as:

$$\boldsymbol{x}''(t) = (\boldsymbol{A} + \boldsymbol{D})\boldsymbol{x}'(t) + (\boldsymbol{BC} - \boldsymbol{AD})\boldsymbol{x}(t) - \boldsymbol{Dm} - \sigma \boldsymbol{D}\boldsymbol{\eta}(t) + \sigma \boldsymbol{\eta}'(t) \tag{42}$$

If we apply the following substitutions from Equations 27 and 17:

- $\boldsymbol{x}(t) \leftarrow \boldsymbol{r}(t)$,
- $\boldsymbol{A} \leftarrow -\sigma_r^2 \Delta t \boldsymbol{\Sigma}^{-1}$,
- $\boldsymbol{B} \leftarrow -\boldsymbol{I}$,
- $\boldsymbol{m} \leftarrow \sigma_r^2 \Delta t \boldsymbol{\Sigma}_t^{-1} \boldsymbol{\mu}$,
- $\boldsymbol{C} \leftarrow -\frac{1}{\tau_a}\boldsymbol{I}$,
- $\boldsymbol{D} \leftarrow \frac{b_a}{\tau_a}\boldsymbol{I}$,

then $\boldsymbol{r}''(t)$ is:

$$\boldsymbol{r}''(t) = \left(\frac{b_a}{\tau_a}\boldsymbol{I} - \sigma_r^2 \Delta t \boldsymbol{\Sigma}^{-1}\right)\boldsymbol{r}'(t)$$

$$+ \left(\frac{1}{\tau_a}\boldsymbol{I} + \frac{b_a}{\tau_a}\sigma_r^2 \Delta t \boldsymbol{\Sigma}^{-1}\right)\boldsymbol{r}(t) \tag{43}$$

$$- \frac{b_a}{\tau_a}\sigma_r^2 \Delta t \boldsymbol{\Sigma}^{-1}\boldsymbol{\mu} - \sigma \frac{b_a}{\tau_a}\boldsymbol{\eta}(t) + \sigma \boldsymbol{\eta}'(t)$$

Recall that, if $\boldsymbol{r}(t)$ follows a stationary Gaussian distribution, then $\nabla_{\boldsymbol{r}(t)} \log p(\boldsymbol{r}(t)) = \boldsymbol{\Sigma}^{-1}(-\boldsymbol{r}(t) + \boldsymbol{\mu})$ (Equation 27), and therefore $\frac{d^2}{d\boldsymbol{r}(t)^2} \log p(\boldsymbol{r}(t)) = -\boldsymbol{\Sigma}^{-1}$. Then,

$$
\begin{aligned}
\boldsymbol{r}''(t) = {} & \left( \frac{b_a}{\tau_a} \boldsymbol{I} + \sigma_r^2 \Delta t \frac{d^2}{d\boldsymbol{r}(t)^2} \log p(\boldsymbol{r}(t)) \right) \boldsymbol{r}'(t) \\
& - \frac{b_a}{\tau_a} \sigma_r^2 \Delta t \, \nabla_{\boldsymbol{r}(t)} \log p(\boldsymbol{r}(t)) + \frac{1}{\tau_a} \boldsymbol{r}(t) \\
& - \sigma \frac{b_a}{\tau_a} \boldsymbol{\eta}(t) + \sigma \boldsymbol{\eta}'(t)
\end{aligned}
\tag{44}
$$

# F    ADDITIONAL RESULTS

Here we present additional results or figures that supplement those of the main text.

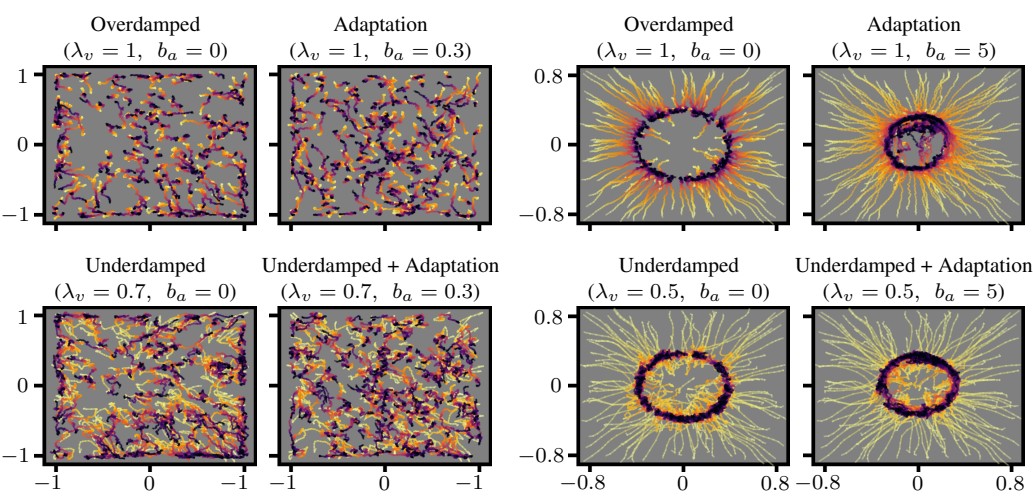

Figure 8:  Example decoded 2D replay trajectories in unbiased (left) and biased (right) rat tasks. Like in Figure 3, trajectories get darker over time.

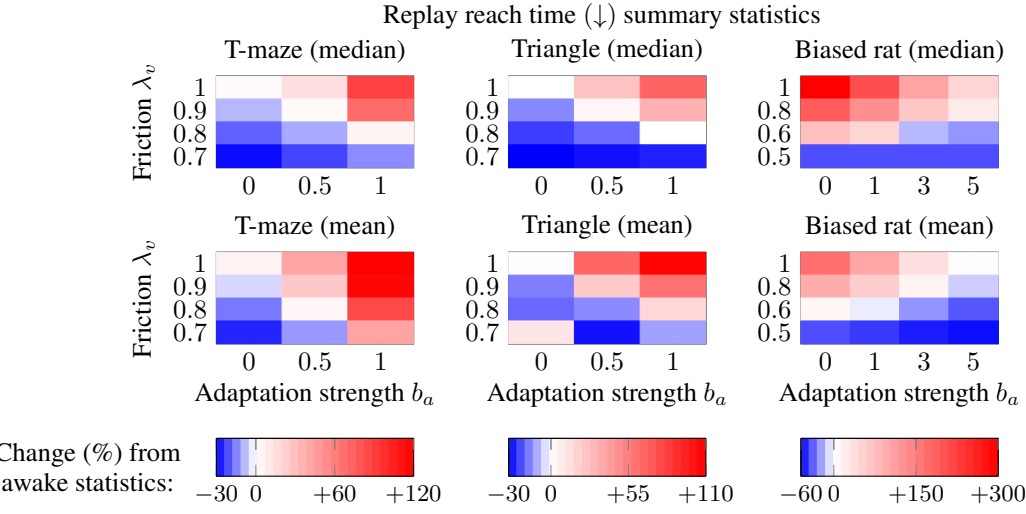

Figure 9:  This is another version of Figure 5, but now with mean reach times also shown.

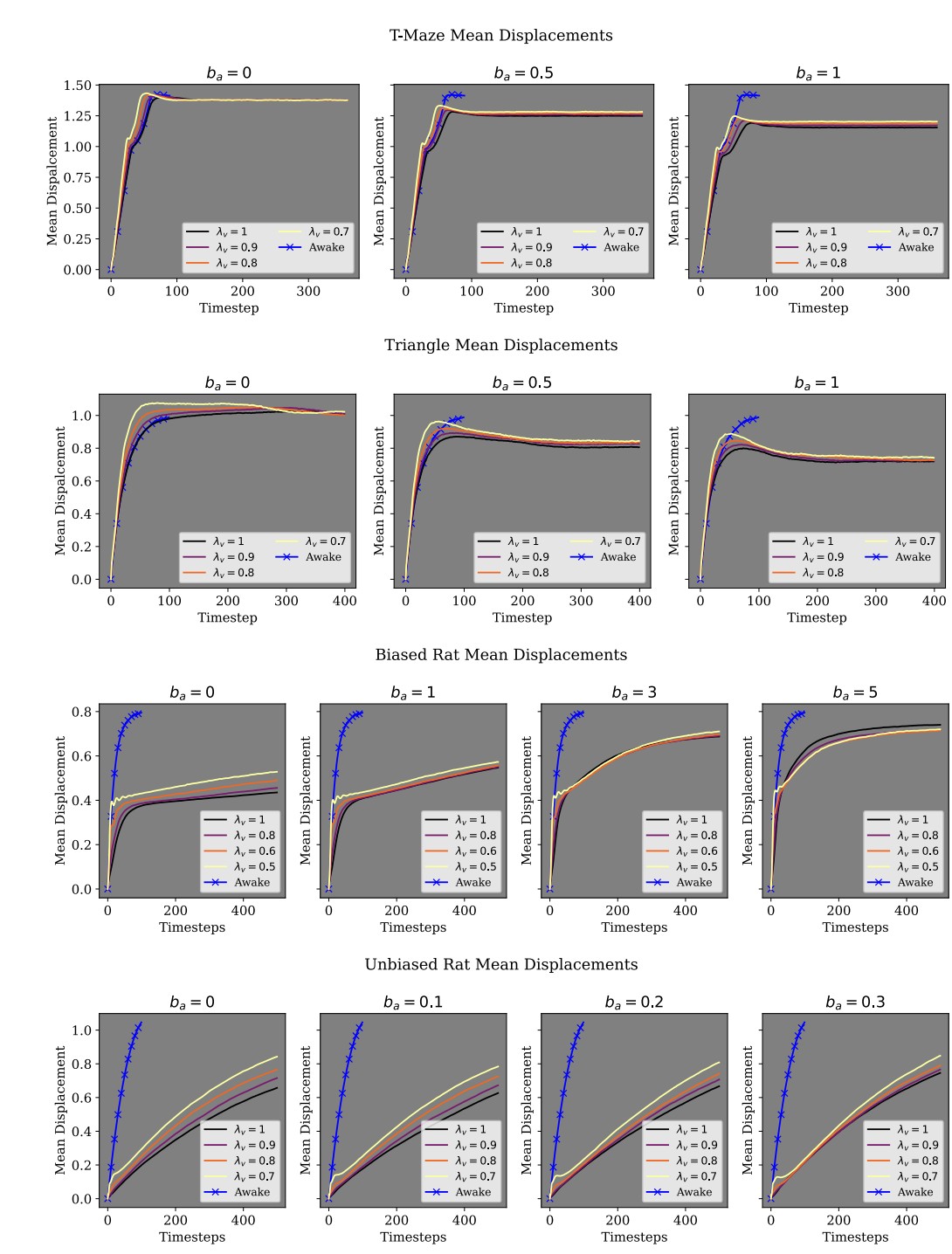

Figure 10: **Underdampening** ($\lambda_v < 1$) **increases mean displacement of replay trajectories, especially at early timesteps.** Similarly to McNamee et al. (2021), here we analyze the *mean displacement*, or distance, of replay trajectories from their starting points as a function of time. For reference, mean displacement over time is also plotted for awake trajectories. In exploration experiments, we simulate replay for $4\times$ the duration of awake trajectories. Higher mean displacements over time generally correlate with increased path length and exploration. Note that awake activity is always calculated with $b_a = 0, \lambda_v = 1$, hence why awake statistics are the same across plots within the same task.

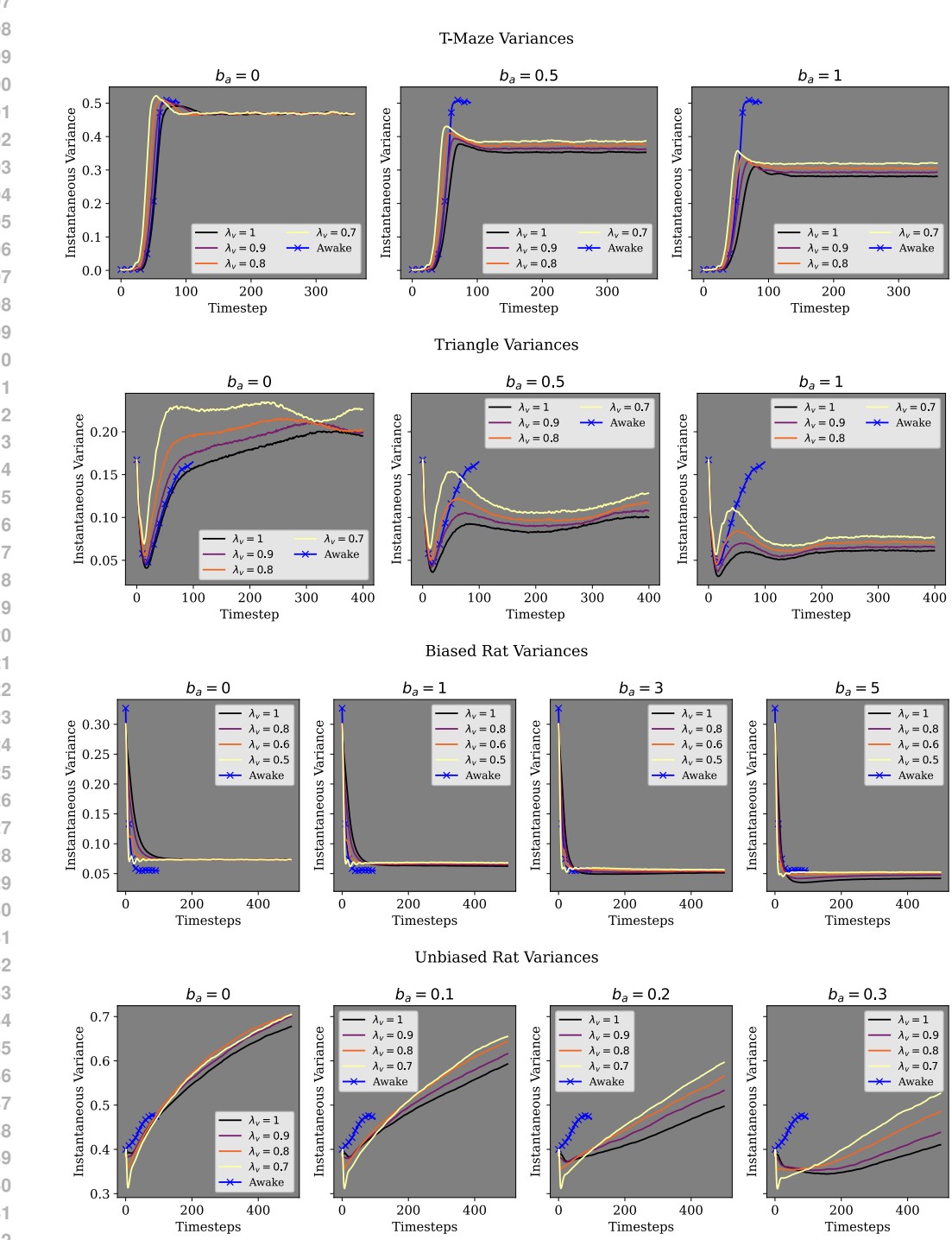

Figure 11: **Underdampening ($\lambda_v < 1$) increases replay trajectory variances.** Like in Figure 10, here we analyze the variance of trajectories at each timestep. Trajectories are 2D, so the variances plotted above are simply the averages of the variances along each coordinate. Underdampening increasing variance is another confirmation that underdampening complements adaptation-induced exploration.

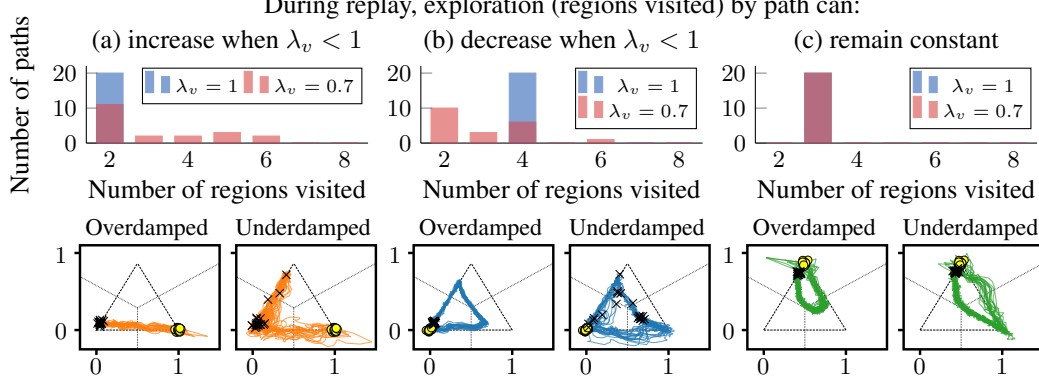

Figure 12: This is another version of Figure 7, but this time with the specific distributions of regions visited shown as bar plots above pairs of replay trajectory sets. Recall that replay paths start at yellow circles and end at $\times$ symbols. In (a), underdampening clearly increases regions visited for several trajectories: instead of just going from $(1, 0)$ to $(0, 0)$, replay paths also continue onwards to traverse between $(0, 0)$ and $(1/2, \sqrt{3}/2)$. Meanwhile in (b), underdampening decreases regions visited for many replay trajectories: instead of traversing the whole triangle (4 regions visited), many replay paths stop in the region defined by $(1, 0)$. As for (c), underdampening did not affect regions visited since all paths started at $(1/2, \sqrt{3}/2)$, visited $(1, 0)$, and then returned to $(1/2, \sqrt{3}/2)$.

# G   RESULTS WITH TANH ACTIVATIONS

Throughout all other parts of the text, we have used piecewise-linear (linear or leaky/non-leaky ReLU) activation functions, as defended in Appendix A.2. For completeness, here we reproduce some key findings of our work with all hyperparameters kept the same, except that RNNs now use tanh activation functions instead of piecewise-linear ones. Replay from tanh RNNs visited far fewer regions than replay from their piecewise-linear counterparts, so a reproduction of Figure 7 is omitted.

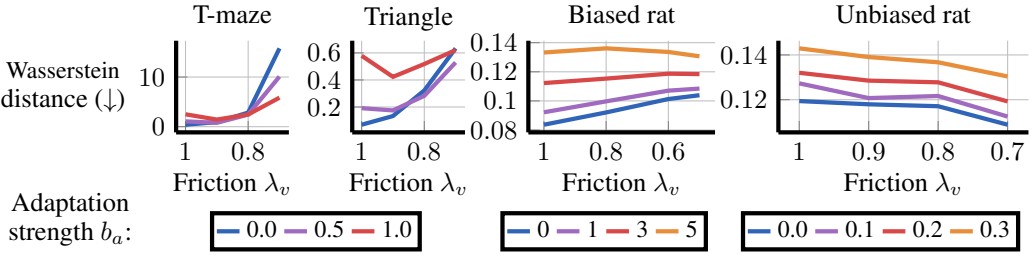

Figure 13: **Mild underdampening can improve replay fidelity in the presence of adaptation in tanh RNNs.** This is another version of Figure 4, for T-maze and triangle tasks, but with tanh RNNs.

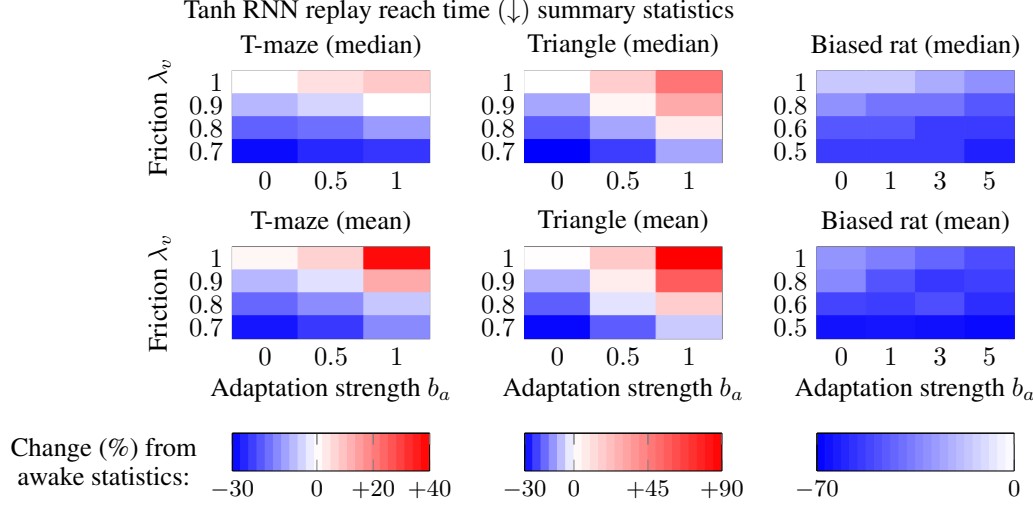

Figure 14: **Underdampening increases speed in tanh RNN replay.** This is another version of Figure 9 for T-maze and triangle tasks, but now with tanh RNNs.

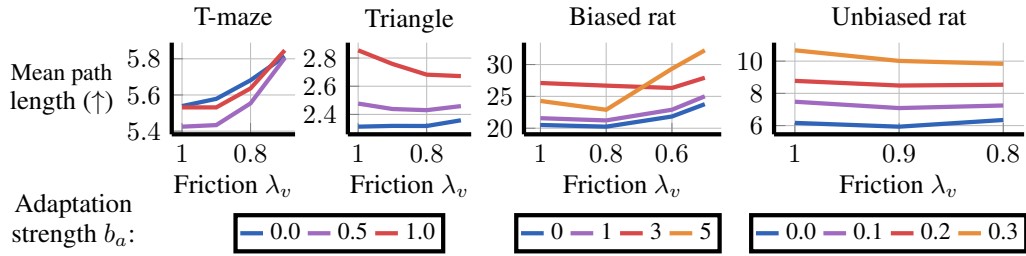

Figure 15: **In tanh RNNs, underdampening may increase or decrease exploration via path length.** This is an alternate version of Figure 6 but with tanh RNNs. Underdampening has mixed effects on tanh RNN replay path length because of tanh saturation.

