# OpenReview forum: "Leakage and Second-Order Dynamics Improve Hippocampal RNN Replay"
_ICLR.cc/2026/Conference — Submitted to ICLR 2026_

### Official Review · Reviewer_FE8C · 2025-10-30

**Soundness:** 2
**Presentation:** 2
**Contribution:** 2
**Rating:** 4
**Confidence:** 2

**Summary:**

This paper builds on existing theory of replay in path-integrating RNNs to provide mechanisms that explain phenomena known to exist in vivo: temporal compression and exploration.

Their first analysis is refuting an assumption from prior work, which assumes that p(r(t)) is stationary, and reveals the role of leakage in helping path integration.

The second analysis proposes two mechanisms (underdampening and adaptation) to explain phenomena of replay observed in vivo, related to the speed of replay: one slows it down and the other one quickens it.

**Strengths:**

S1. A principled, quantitative approach to the phenomenon of replay.

S2. Great related works.

S3. The figures from the experiments are very clear and comprehensive. The experiments themselves seem sound to illustrate and support the theoretical results.

**Weaknesses:**

W1. The motivation behind the proposed mechanisms is difficult to follow. Maybe it is clearer to a neuroscience audience, but for an ICLR audience it needs to be explained. The authors refute the stationarity of p(r(t)) but I had trouble following *why* this assumption must be violated for their framework to hold.

Similarly, why this choice of underdamping and adaptation mechanisms? While they indeed modulate the temporal dynamics (e.g., the speed of replay), the paper does not convincingly argue (or I missed it) why these mechanisms were chosen over other plausible alternatives.

W2. The intended scope of the framework is a bit unclear. Is the work primarily aimed to model biological neural circuits (in which case novel, testable predictions should be articulated, and claims should be supported by biological experiments) or to propose computational mechanisms relevant for RNNs (in which case implications for model training or performance should be discussed)? Clarifying this distinction would help position the contributions.

W3. Section 2 is hard to follow. The sequence of Definition, Assumption, Lemma, Theorem is a bit abrupt and difficult to parse. It could benefit from a higher-level narrative or schematic overview or illustrative toy examples. As written, it's hard to follow how each statement supports the main argument.

W4. The claim that the proposed mechanisms “improve hippocampal RNN replay” is currently unsubstantiated. The paper presents qualitative examples but lacks quantitative metrics or statistical comparisons with experimental data.

The use of the term “hippocampal” seems too specific given the lack of biological validation. Replace with "path integration" o?Using more neutral terminology would avoid overinterpretation of the biological relevance.

**Questions:**

How many RNNs (different seeds) were trained each time?

The definition of RNN in eqn (4) is very generic. What function f is used in practice (leaky current? leaky firing rate?)  and what is the biological faithfulness of that choice?

---

> ### Author Response · Authors · 2025-11-21
>
> We thank the reviewer for their time and their critique of our work. We appreciate that the reviewer recognizes the principled, quantitative nature of our work and comprehensive review and experimental results. We are committed to addressing the reviewer’s concerns and provide our responses below.
>
> > W1.1. The authors refute the stationarity of p(r(t)) but I had trouble following why this assumption must be violated for their framework to hold.
>
> We apologize, but there may be a misunderstanding here. The non-stationarity of $p(r(t))$ is _not_ a requirement for our results to hold. In fact, if $p(r(t))$ (and thus, $p(s(t))$) is stationary, then Langevin sampling convergence guarantees hold, meaning that replay distributions are essentially guaranteed to capture $p(s(t))$. Moreover, stationarity should ensure that underdamped Langevin sampling improves upon the fidelity of $p(r(t))$ compared to $p(s(t))$. In summary, stationarity would be a very convenient property for replay-generating networks.
>
> However, we show throughout the text that stationarity is a property of path trajectories that does not hold for even basic distributions like Markov Gaussian processes (Figure 1). This non-stationarity makes estimating the score function of awake trajectories challenging (Section 3), and is why we have to use systems with memory (i.e., RNNs) in order to produce replay as the score function is time-varying (on the contrary, Langevin sampling from a stationary distribution can be done through repeated calls of a feedforward network, analogous to a diffusion model without noise scheduling).
>
> Furthermore, because of non-stationarity, classic techniques like underdamped Langevin sampling are not guaranteed to improve sampling fidelity (in Figure 4, some Wasserstein distances increase as underdampening friction is decreased, especially without adaptation). Lastly, we note that our non-stationary processes in fact converge to stationary distributions: trajectories start moving towards their goals (non-stationary), and then rest there (stationary). For more discussion of stationarity, please refer to Appendix A.6 of the existing submission, “Deviations from Traditional Langevin Sampling”.
>
> > W1.2 Similarly, why this choice of underdamping and adaptation mechanisms?
>
> We thank the reviewer for the question. We explored this particular adaptation mechanism because it has been explored in prior work [1], and the underdampening mechanism through momentum is simple, general, and akin to various proposals for accelerated sampling in the theoretical neuroscience literature [2–5]. While there may be other forms of adaptation and underdampening, these mechanisms are grounded in prior work and designed to be simple and broadly generalizable. If the reviewer has specific suggestions for alternatives to explore, we would be happy to provide additional context or discussion.
>
> In more detail, we draw attention to Section 2.1 (“Langevin Dynamics”) and the “Neural adaptation” paragraph of Section 2.3 (“Existing Methods of Biasing Replay in RNNs”). The reason for investigating underdampening in the first place follows from the perspective established by Krishna et al. [6] wherein replay can be considered Langevin sampling: if that holds, then it is logical to see whether variants of Langevin sampling can improve replay, and whether such variants could have biological analogues. We keep our mechanism generic so that it is broadly generalizable and captures the key question of how momentum in sampling affects replay statistics, rather than focusing on the fine details of mapping these mechanisms onto more detailed models with specific implementations such as E/I circuits. Important future work can explore this direction in line with certain prior work [2–5], and also work on comparing our results to biology through more targeted experimentation. There has been some recent work showing that humans sample with momentum [7] or that hippocampal sharp wave ripples in rats show momentum effects [8]. Experiments could focus on identifying what processes underlie this momentum or temporal compression, e.g., if short-term facilitation underlies momentum, lesioning for it could result in different replay statistics.
>
> As for adaptation, negative feedback is a canonical neural computation with a myriad of functional implications, and has been previously examined in path-integrating networks and continuous attractor networks in other works. Levenstein et al. [1] most recently applied adaptation for replay in trained RNNs, which we investigate further. For further discussion and related works, please refer to Section 2.3. Our objective was to reconcile the differences and bridge the gap between works such as Krishna et al. [6] which did not use adaptation and Levenstein et al. [1] which did, to understand the influence of such mechanisms on replay statistics. We would be happy to incorporate additional mechanisms if the reviewer has specific suggestions.

---

> > ### Author Response · Authors · 2025-11-21
> >
> > > W2. Is the work primarily aimed to model biological neural circuits (in which case novel, testable predictions should be articulated, and claims should be supported by biological experiments) or to propose computational mechanisms relevant for RNNs (in which case implications for model training or performance should be discussed)?
> >
> > Our paper is theoretical, and seeks to understand how replay from path-integrating RNNs (which model an important function of navigational systems like the hippocampus and entorhinal cortex) is affected by design parameters like leakage, adaptation, and underdamped sampling. Model training is always done without second-order modifiers like adaptation and underdampening, and we do evaluate performance with respect to leakage.
> >
> > Our focus is on coming up with an understanding of how the “performance” or statistical properties of replay vary as a function of second-order modifiers (adaptation and underdampening). We are motivated to explore these underdampening and adaptation mechanisms to better match observations of replay in biology: we propose that momentum in sampling could underlie observed temporal compression of replay events, and in line with prior work, study adaptation’s effect on the diversity and length of replay sequences. We also comprehensively analyze a tradeoff between underdampening and adaptation, which to our knowledge has not been carried out before.
> >
> > Hippocampal recordings and data analysis in the context of replay activity are notoriously difficult. Given these difficulties, we argue that theoretical results are necessary to provide directions for experimentalists to pursue, in line with several theoretical works on the hippocampal formation [1,6,9–11]; we also believe that more targeted experimentation would be necessary to compare our results to those from biological experiments. Overall:
> > * Comparing neural data and artificial network activations is highly non-trivial. This is especially the case with replay. Replay events are sparse and embedded in a sea of complex, noisy dynamics. Detecting replay events in vivo is itself a research challenge, so comparing with replay data would be outside the scope of our work, especially considering that there would be no clear mapping from an artificial neural network to a biological one. Future work could evaluate our predictions in biological networks through targeted experiments.
> > * We also argue that our theoretical work has value in neuroscience [12], where biological experiments can be expensive and difficult. Our work should be seen as understanding key components of and improving upon existing theoretical results to account for experimental observations in replay, such as temporal compression and exploration. Lying at the intersection of mechanistic and normative modeling (i.e., explaining how and why a phenomenon might arise), our theoretical work proposes mechanisms that could better account for replay in biological circuits, and also outlines paths for future experiments on replay, such as probing the connection between short-term facilitation and temporal compression of replay events.
> > * We try to address a lack of biological data by working with synthetic models that mimic it, such as RatInABox [13]. We simulated synthetic rat trajectories and place cells, then trained an RNN to path-integrate place cell activity, and then decoded place cell activity during replay to analyze 2D synthetic rat replay trajectories (Figures 4–11).
> >
> > We hope these points convince the reviewer of the value of theoretical works like ours in neuroscience, and hope they will reconsider their assessment.

---

> > > ### Author Response · Authors · 2025-11-21
> > >
> > > > W3. Section 2 is hard to follow. The sequence of Definition, Assumption, Lemma, Theorem is a bit abrupt and difficult to parse. It could benefit from a higher-level narrative or schematic overview or illustrative toy examples. As written, it's hard to follow how each statement supports the main argument.
> > >
> > > We apologize for any confusion that our style of presenting theoretical background and results may have caused. However, there are a few important points to keep in mind with regards to our organization of mathematical results:
> > > * Section 2 is _entirely background_. Section 2.2 is simply meant to be a concise and clear summary of the theoretical results from Krishna et al. [6], with enough information to be thorough and clear with regards to how RNNs perform Langevin sampling and what assumptions are necessary to come to that conclusion. This goal requires reducing Krishna et al.’s work to its most essential definitions, assumptions, and results, with little room for extra content. For further explanation of their work, we invite the reader to consult Krishna et al. [6].
> > > * Secondly, our paper is largely theoretical in nature, and as such requires a certain level of mathematical clarity. Organizing the paper alternatively would be unconventional and unclear as a theoretical work. As we will show below, we have added a high-level intuition/understanding guide for this section in Appendix B.
> > > * We do not just provide the background in Section 2.2 without context. Throughout the introduction, there is a higher-level narrative concerning how RNNs can perform replay via Langevin sampling. We make sure to place Figure 1 at the beginning of the paper to give the reader some idea of what replay could look like for a basic Gaussian distribution, and we preface Section 2.2 with Section 2.1, which defines Langevin sampling, and a paragraph which explains path-integration and how it is relevant to Langevin sampling via the score function of the input distribution.
> > >
> > > We are committed to ensuring that our work is accessible to a broader audience, and so in our current revision we have provided additional context, as well as a new Appendix B with a high-level description and intuitions to understand Section 2.2.
> > >
> > > _We reproduce excerpts below and hope this alleviates the reviewer’s concerns about the background:_
> > >
> > > * Our work examines how trained noisy RNNs can act as generative models in the absence of inputs, which prior work including Krishna et al. (2024) consider to be models of replay in neuronal circuits.
> > > * Accordingly, we define Langevin dynamics, with (Equation 1) and without (Equations 2 and 3) momentum, as an iterative process by which an agent could attempt to sample from an unknown distribution $p(x)$ using only knowledge of its score function $\nabla \log p(x)$, which can be learned from noisy observations drawn from the distribution. The following points provide more detail on the sampling process and why Langevin dynamics arise:
> > >   * Biological networks such as those underlying navigation must accurately estimate environmental state variables such as position from observations of self-motion, even in the presence of intrinsic noise.
> > >   * This noise induces a distribution over network states, i.e., activity, however, it is unknown exactly what the true noise distribution is.
> > >   * Accurate state estimation would thus require the network to optimally remove intrinsic noise without access to the exact parameters of the noise distribution, but given noisy network states.
> > >   * This optimal solution can be accomplished by learning the score function of noisy network states and using it to denoise network states over time as additional inputs and noise are presented (Miyasawa, 1961).
> > >   * When the network receives no inputs, its noise-driven dynamics still attempt to remove intrinsic noise, but the learned score function is for the distribution of waking task-like activity. This causes the network states to resemble actual samples drawn from the distribution over network states in the presence of inputs. The theoretical results show that the network dynamics during this quiescent state carry out sampling by Langevin dynamics, i.e., using the score function of the network's activity distribution during task performance to yield quiescent network activity that resembles waking activity.

---

> > > > ### Author Response · Authors · 2025-11-21
> > > >
> > > > _Excerpt of Appendix B (contd.):_
> > > >
> > > > * Biological neural circuits like the hippocampus can navigate through environments when awake, and recall navigatory episodes during rest. Accordingly, we summarize the proof from Krishna et al. (2024) that a noisy RNN trained on a biologically relevant navigation task like path-integration can act as a generative model.
> > > >   * Definitions 1 and 2 simply express that our RNNs have nonlinear dynamics and internal noise, and are trained to estimate a signal $\boldsymbol{s}(t)$ by integrating observations of $\boldsymbol{s}'(t)$. This phenomenon, known as path-integration, underlies navigation and is used to estimate one's position from self-motion cues.
> > > >   * Assumption 1 asserts that RNN dynamics can be decomposed into additive components.
> > > >   * Assumption 2 simply states that the optimal RNN dynamics would lead to accurate path-integration, such that the conditional probability distribution over network states (activity) given environmental states (position) would be Gaussian and identical to the additive noise in our networks. That is, the variability over network states would only be due to intrinsic noise in the system and have the same statistics.
> > > >   * Finally, Assumption 3 assumes that the RNN dynamics are greedily optimal for path-integration. Greedy optimization is a sensible way of partitioning effort across time in path-integration: the network does the best that it can at each timestep, assuming that at each previous timestep the best possible job has been done.
> > > >   * Theorem 1 states that path-integrative RNN dynamics will use the score function of waking activity to remove intrinsic noise from the system and use inputs to update state estimates. This is a two-step greedily optimal solution for path integration in the presence of intrinsic noise: first, intrinsic noise must be removed, following which the inputs must be used to update the estimate of the current environmental state.
> > > >   * Theorem 2 shows that these dynamics will result in Langevin sampling from the distribution of waking activity in the absence of any inputs, i.e., analogous to sleep. This means that the quiescent network dynamics sample neural states from the same distribution as waking, task-like activity, and can thus represent sequences like those during awake task performance, i.e., leading to replay.
> > > >
> > > > * Finally, we define existing methods of modulating RNN activity or training that affect RNN replay distributions: adaptation (negative feedback), which encourages diversity in replay paths, and masked training, which encourages coherence in replay.

---

> ### Author Response · Authors · 2025-11-21
>
> > W4. The claim that the proposed mechanisms “improve hippocampal RNN replay” is currently unsubstantiated. The paper presents qualitative examples but lacks quantitative metrics or statistical comparisons with experimental data.
>
> We refer to our previous response to W2 and also our [response to common questions](https://openreview.net/forum?id=kGNtM8NteM&noteId=oGReodDPn1) for a discussion on why comparison to experimental data is non-trivial (in fact, datasets that allow us to specifically test for these mechanisms do not currently exist, to our knowledge) and how our work should be seen as a proposal for targeted experimental verification.
>
> We would also like to note that we have quantified several characteristics of replay trajectories and compared them numerically to those of awake trajectories: in Figure 4, we compute Wasserstein distances (a very common metric for generative models, which our RNNs are); in Figures 5 and 9, we estimate the summary statistics of the speed of replay; in Figure 6, we calculate the average length of replay trajectories; in Figures 7 and 12, we calculate numerical measures of exploration (note that exploration has no existing standard metrics); and in Figures 10 and 11, we calculate average mean displacements (like McNamee et al. [14]) and variances of replay trajectories over time.
>
> Finally, given that the task considered in this work is path-integration underlying navigation, which is implemented by the entorhinal cortex in conjunction with the hippocampus, our theoretical work is highly relevant to the hippocampal formation, just like prior work [1,6,9–11]. We argue that comparisons to biological data are not necessary to claim relevance to the hippocampus, rather the tasks considered and the systems implicated in those tasks are part of the hippocampal formulation and our work should be considered as a model of replay in the same.
>
> > Q1. How many RNNs (different seeds) were trained each time?
>
> We apologize that this was not mentioned, we had erroneously forgotten to include this in the paper. We use 5 seeds, which we mention now in section A.1.2.
>
> > Q2. The definition of RNN in eqn (4) is very generic. What function f is used in practice (leaky current? leaky firing rate?)  and what is the biological faithfulness of that choice?
>
> Thank you for the question. Equation 4 is a restatement from Krishna et al. [6], where the aim is to keep the RNN as generic as possible with $f$ being sufficiently expressive as stated in their paper. Specifically, we use simple, vanilla RNNs, with Leaky ReLU activations in 2D tasks, and ReLU activations in rat tasks. We defend these activation functions in Section A.2.1, “Activation Function”, of the manuscript. We also show results with $\mathrm{tanh}$ activations in the new Appendix G, where the results largely remain consistent.
>
> In general, we argue that ReLU RNNs offer better biological realism leading to their prevalence in all prior work we have considered, as outlined in Appendix A.2.1.  Compared to RNNs with multiplicative gating interactions (e.g., GRUs and LSTMs) or activations such as $\mathrm{tanh}$, ReLU RNNs are more biologically plausible insofar as they can be easily interpreted as the nonnegative, sparse firing rates of spiking networks. This is a key reason (among others listed in Appendix A.2.1) that motivates the use of ReLU networks in our work.
>
> ---
>
> Overall, we thank the reviewer for their valuable comments. We hope that our new revision addresses the issues raised by the reviewer, and also that our comments have alleviated their other concerns.  If so, we would kindly request the reviewer to reconsider their assessment and potentially recommend acceptance. We are happy to address any other concerns from the reviewer during the discussion phase and are committed to improving our work.

---

> > ### Author Response · Authors · 2025-11-21
> >
> > **References:**
> >
> > 1. Levenstein, Daniel, et al. "Sequential predictive learning is a unifying theory for hippocampal representation and replay." bioRxiv (2024): 2024-04.
> > 2. Hennequin, Guillaume, Laurence Aitchison, and Máté Lengyel. "Fast sampling-based inference in balanced neuronal networks." Advances in neural information processing systems27 (2014).
> > 3. Masset, Paul, et al. "Natural gradient enables fast sampling in spiking neural networks." Advances in neural information processing systems 35 (2022): 22018-22034.
> > 4. Dong, Xingsi, and Si Wu. "Neural sampling in hierarchical exponential-family energy-based models." Advances in Neural Information Processing Systems 36 (2023): 78593-78606.
> > 5. Aitchison, Laurence, and Máté Lengyel. "The Hamiltonian brain: Efficient probabilistic inference with excitatory-inhibitory neural circuit dynamics." PLoS computational biology 12.12 (2016): e1005186.
> > 6. Krishna, Nanda H., et al. "Sufficient conditions for offline reactivation in recurrent neural networks." The Twelfth International Conference on Learning Representations.
> > 7. Castillo, Lucas, et al. "Explaining the flaws in human random generation as local sampling with momentum." PLOS Computational Biology 20.1 (2024): e1011739.
> > 8. Krause, Emma L., and Jan Drugowitsch. "A large majority of awake hippocampal sharp-wave ripples feature spatial trajectories with momentum." Neuron 110.4 (2022): 722-733.
> > 9. Bredenberg, Colin, et al. "Impression learning: Online representation learning with synaptic plasticity." Advances in neural information processing systems 34 (2021): 11717-11729.
> > 10. George, Tom M., et al. "A generative model of the hippocampal formation trained with theta driven local learning rules." Advances in Neural Information Processing Systems 36 (2023): 1644-1658.
> > 11. Sorscher, Ben, et al. "A unified theory for the origin of grid cells through the lens of pattern formation." Advances in neural information processing systems 32 (2019).
> > 12. Levenstein, Daniel, et al. "On the role of theory and modeling in neuroscience." Journal of Neuroscience 43.7 (2023): 1074-1088.
> > 13. George, Tom M., et al. "RatInABox, a toolkit for modelling locomotion and neuronal activity in continuous environments." Elife 13 (2024): e85274.
> > 14. McNamee, Daniel C., et al. "Flexible modulation of sequence generation in the entorhinal–hippocampal system." Nature neuroscience 24.6 (2021): 851-862.

---

### Official Review · Reviewer_Qmcp · 2025-10-31

**Soundness:** 3
**Presentation:** 2
**Contribution:** 3
**Rating:** 6
**Confidence:** 4

**Summary:**

This paper aims to improve the understanding of replay in RNNs from the perspective of stochastic sampling and dynamical systems theory. The authors re-examine how specific network mechanisms, such as leakage, adaptation, and momentum, shape the resulting replay dynamics. They first show that the score function driving replay is time-varying and difficult to estimate directly, motivating the inclusion of hidden-state leakage as a stabilizing inductive bias that improves both training and replay quality. Next, they show that adaptation, modeled as negative feedback, encourages exploration in the RNN’s internally generated activity but results in slower replay. To address this, the authors introduce hidden-state momentum, connecting it to underdamped Langevin dynamics, which yields faster and more temporally compressed replay that more closely mirrors hippocampal replay observed in the brain. Experiments in simulated 2D navigation and synthetic place-cell environments show that combining adaptation with underdamped dynamics produces replay that is both diverse and efficiently time-compressed.

**Strengths:**

This paper provides a deeper understanding of replay dynamics in RNNs under a set of strong but well-motivated assumptions. It effectively bridges neuroscience and machine learning, which machine learning draws inspiration from. They are seemingly the first to introduce an RNN framework that exhibits temporally compressed replay, resembling replay observed in the hippocampus. The experiments, though conducted in simplified environments, are thoughtfully designed and demonstrate how mechanisms such as leakage, adaptation, and underdamped dynamics influence replay.

**Weaknesses:**

While this paper helps further the understanding in the study of replay dynamics, several aspects could be improved for clarity and empirical depth. Some figures. such as Figure 1, could more clearly distinguish functions such as s(t) and r(t) to improve readability. The discussion linking the underdamped mechanism to short-term facilitation is intriguing but remains speculative without experimental validation or quantitative comparison to biological data. Additionally, the claim that the proposed mechanisms improve replay is supported primarily by qualitative evidence; including more quantitative analyses would help strengthen the paper’s conclusions.

**Questions:**

Can similar conclusions be made for different architectures like gated RNNs?

---

> ### Author Response · Authors · 2025-11-21
>
> We would like to thank the reviewer for their time and their positive review of our work. We address the points raised by the reviewer below.
>
> > W1. Some figures. such as Figure 1, could more clearly distinguish functions such as s(t) and r(t) to improve readability.
>
> We apologize for any lack of clarity in Figure 1, however, it is unclear to us how Figure 1 currently lacks readability. We are open to specific suggestions from the reviewer on improving this figure. We checked the current version of Figure 1 in a monochrome environment, and verified that $s(t)$, $r(t)$, and the standard deviations thereof are clearly distinguishable even without any color information. Note that $s(t)$ and $r(t)$ are supposed to be similar, since $r(t)$ should approximate $s(t)$ in replay from a successful path-integration network. Furthermore, we also ensured that the existing plot headers and captions completely and clearly described $s(t)$ and $r(t)$. We are happy to make any specific modifications the reviewer suggests in order to improve the figure.
>
> > W2. The discussion linking the underdamped mechanism to short-term facilitation is intriguing but remains speculative without experimental validation or quantitative comparison to biological data. Additionally, the claim that the proposed mechanisms improve replay is supported primarily by qualitative evidence; including more quantitative analyses would help strengthen the paper’s conclusions.
>
> While we acknowledge that we do not compare to biological data in our work, we would like to respectfully push back on the claim that our results are supported primarily by qualitative evidence and not quantitative evidence. While we provide a response under the “On quantitative evidence” section of our [response to common questions](https://openreview.net/forum?id=kGNtM8NteM&noteId=oGReodDPn1), we would like to reiterate our points here.
>
> We argue that given the difficulty of experiments on replay, our theoretical results should be seen as providing directions for experimentalists to pursue; we also believe that more targeted experimentation would be necessary to compare our results to those from biological experiments. Overall:
> * Comparing _neural data and artificial network activations is non-trivial_. This is especially the case with replay. Replay events are sparse and embedded in a _sea of complex, noisy dynamics_. Detecting replay events in vivo is itself a _research challenge_, so comparing with replay data would be outside the scope of our work, especially considering that there would be no clear mapping from an artificial neural network to a biological one. Future work should evaluate our predictions in biological networks through _targeted experiments_.
> * We also argue that our _theoretical work has value in neuroscience_ [1], where biological experiments can be expensive and difficult. Our work should be seen as _understanding key components of and improving upon existing theoretical results to account for experimental observations in replay_, such as temporal compression and exploration. Lying at the intersection of mechanistic and normative modeling (i.e., explaining how and why a phenomenon might arise), our theoretical work proposes mechanisms that could better account for replay in biological circuits, and also outlines paths for future experiments on replay, such as _probing the connection between short-term facilitation and temporal compression_ of replay events.
> * We try to address a lack of biological data by working with _synthetic models that mimic_ it, such as RatInABox [2]. We simulated synthetic rat trajectories and place cells, then trained an RNN to path-integrate place cell activity, and then decoded place cell activity during replay to analyze 2D synthetic rat replay trajectories (Figures 4–11).
> * We _quantified several characteristics of replay trajectories_ and _compared them numerically to those of waking trajectories_: in Figure 4, we compute _Wasserstein distances_ (a very common metric for generative models, which our RNNs are); in Figures 5 and 9, we estimate the _summary statistics of the speed of replay_; in Figure 6, we calculate the _average length of replay trajectories_; in Figures 7 and 12, we calculate _numerical measures of exploration_ (note that exploration has no existing standard metrics); and in Figures 10 and 11, we calculate _average mean displacements_ (like McNamee et al. [3]) and _variances of replay trajectories_ over time.
>
> > W3. Can similar conclusions be made for different architectures like gated RNNs?
>
> We thank the reviewer for this important question. We have provided a response to this question in our [response to common questions](https://openreview.net/forum?id=kGNtM8NteM&noteId=oGReodDPn1). In short, the overall results do not change drastically in the presence of other activation functions such as $\mathrm{tanh}$, as explained by our new results in Appendix G.

---

> ### Author Response · Authors · 2025-11-21
>
> Overall, we thank the reviewer again for their positive review and for engaging with our work. We hope that our response clarifies the reviewer’s concerns, and kindly request the reviewer to consider supporting our work more strongly if so. In particular, we hope that our clarifications on the thorough quantitative experiments in our paper will improve the reviewer’s opinion of our work. We are also happy to address any further concerns from the reviewer during the discussion phase.
>
> **References**
>
> 1. Levenstein, Daniel, et al. "Sequential predictive learning is a unifying theory for hippocampal representation and replay." bioRxiv (2024): 2024-04.
> 2. George, Tom M., et al. "RatInABox, a toolkit for modelling locomotion and neuronal activity in continuous environments." Elife 13 (2024): e85274.
> 3. McNamee, Daniel C., et al. "Flexible modulation of sequence generation in the entorhinal–hippocampal system." Nature neuroscience 24.6 (2021): 851-862.

---

### Official Review · Reviewer_WDVo · 2025-10-31

**Soundness:** 2
**Presentation:** 1
**Contribution:** 3
**Rating:** 4
**Confidence:** 3

**Summary:**

Authors study RNNs in the context of hippocampal replay, in which RNNs are first trained in the presence of task-relevant inputs and later their neural activations are sampled without inputs. Authors show that the score function of the resulting probability distribution for the neural activities is not stationary, which corrects an incorrect assumption in the literature, and incorporate three biologically relevant mechanisms under one common framework.

**Strengths:**

- Authors present a convincing unified theoretical framework of RNN replay that incorporates membrane voltage decay, adaptation, and STSP.

- The paper identifies the invalidity of a key assumption made regularly in the literature.

- Authors explain modulation of replay speed, which to my understanding was not explained with theoretical models before.

**Weaknesses:**

- Presentation-wise, I find the current manuscript very dense. For instance, what does it mean for an RNN to perform Langevin sampling? What is sampling, what is being sampled? A more comprehensive background for the broader audience may be desirable here. Also, how can an RNN sleep? Do the authors refer to running RNNs without any inputs? If so, what are the initial conditions? As a computational neuroscientists not immediately working on these topics, I was at first very confused about the content of the work. For publication in ICLR, which is not a specialized journal, a lot less jargon would have ben desirable.

- My major concern on the results is that authors are rediscovering the fact that discretized RNNs, with discretization time being equal to the neural decay constant $Δ t = \tau$, are actually not a good model of continuous neural dynamics. Allow me to elaborate. The case with no leakage is actually not an accurate approximation to the continuous RNN dynamics $\tau \dot r = -r + f(r(t),u(t))$, where $f(r(t),u(t))$ is the flow map. If we discretize this with Euler's discretization, we arrive at $r(t+Δt) = (1-Δ t/ \tau) r(t) + (1-Δ t/\tau) f(r(t),u(t))$. Now, the line of research that authors are citing simply assumed $Δ t = \tau$, which is certainly not reasonable since the discretization only makes sence if $Δ t \ll \tau$. Now, the authors once again are finding that "leakage" is important for path-integration, which simply means $\tau$ needs to be set properly based on task demands. In other words, scientifically, this simply means one should let the task decide the timescales. There is no "leakage" in continuous systems, but the time scale of the decay $\tau$.

**Questions:**

Can you please address the two weaknesses above?

My overall assessment is as follows: This work is interesting, and does have the novelty and potential impact that ICLR seeks. Some reframing of the results is needed (e.g., leakage...), and presentation needs to be cleaned up significantly. Once these changes are made, I would feel comfortable recommending an acceptance.

---

> ### Author Response · Authors · 2025-11-21
>
> We sincerely thank the reviewer for their time and their critique of our paper. We appreciate that the reviewer recognizes our contributions as interesting, novel, and potentially impactful. We have updated our manuscript in line with comments from the reviewer (changes in forest green) and will address the reviewer’s concerns below.
>
> > W1.1. Presentation-wise, I find the current manuscript very dense. For instance, what does it mean for an RNN to perform Langevin sampling? What is sampling, what is being sampled? A more comprehensive background for the broader audience may be desirable here.
>
> We apologize for any lack of clarity. In our new revision, we have attempted to provide ample explanation and intuition for background concepts and theoretical results. While we had already provided background on Langevin sampling (Section 2.1) and how RNN implementations of it deviate from other widely known models (Appendix A.6), we have expanded the former and other sections with more information, and added a new Appendix B on a detailed, high-level summary of our Background section. We reproduce the main changes below and invite the reviewer to refer to the changes in forest green in the new revision for more details.
>
> _Summarizing the goal of the background section and introducing the problem clearly (Lines 118-126; Section 2: Background)_
>
> In this section, we provide background and summarize results from prior work (Krishna et al., 2024) that has described replay in path-integrating neural circuits as the sampling during quiescence (i.e., the absence of any inputs) of neural activity states from the distribution of waking, task-like activity. _In short, the dynamics of recurrent networks that learn to optimally path-integrate noisy inputs cause the network's states even in the absence of inputs to resemble those attained during actual task performance._ We also provide details on mechanisms used in the literature to improve or bias replay, which we explore the effects of in more detail in this work. For an overview of this section, see Appendix B.
>
> _Introducing Langevin dynamics in the context of RNNs; also clarifying that network states are sampled (Lines 140-146; Section 2.1: Langevin dynamics)_
>
> We consider noisy RNNs that have implicitly learned to implement Langevin sampling of their own fixed distributions of network activity during task performance, when driven by just intrinsic noise and in the absence of inputs. That is, _the activity of the RNNs at each unrolled timestep in the absence of inputs represents a plausible and likely vector of network activity during actual task performance_ in the presence of inputs. In particular, we view such networks in the context of replay, where neural circuits recapitulate task-like activity even during sleep.
>
> _Tying together Langevin dynamics and replay, i.e., introducing contributions of prior work proposing that replay could be implemented through Langevin sampling (Lines 151-156; Section 2.2: Offline Replay in RNNs)_
>
> Krishna et al. (2024) have shown how noisy RNNs trained to path-integrate their inputs implicitly learn statistics that produce Langevin sampling of _their own task-relevant activity_ distribution when no input is given, demonstrating _statistically faithful replay sequences during quiescence_. That is, the _RNNs' activity in the absence of inputs “replays” states from the same distribution as RNN activity with inputs during actual task performance_. This leads to the generation or “replay” of output sequences resembling those that the network sees during training or task performance with inputs.
>
> _Background on sampling and replay (will add in a subsequent revision pending reviewer’s approval)_
>
> Sampling is the process of selecting observations $x$ from a probability distribution $p(x)$. Typically, one does not have access to the exact parameters of $p(x)$ and is given only samples thereof, using which an approximation $q(x)$ is learned. In terms of replay, $x$ is a trajectory of network activity and $p(x)$ is the true distribution of waking (input-driven) activity trajectories. In order to perform optimal path integration in the presence of noise, the network must learn $q(x) \approx p(x)$ in order to compensate for intrinsic noise, and thus $q(x)$ is the distribution of replay (internally/noise-driven) trajectories. For an RNN’s activity to attain values that are likely under $p(x)$, it may implement Langevin sampling from its estimate $q(x)$ of $p(x)$. The network’s dynamics simply follow the gradient of log-likelihood of $q(x)$ at every timestep; in other words, at each timestep, it tries to steer its activity trajectories in high-likelihood directions.

---

> > ### Author Response · Authors · 2025-11-21
> >
> > _Excerpt of Appendix B with high-level intuition_
> >
> > Our work examines how trained noisy RNNs can act as generative models in the absence of inputs, which prior work including Krishna et al. (2024) consider to be models of replay in neuronal circuits. Accordingly, we define Langevin dynamics, with (Equation 1) and without (Equations 2 and 3) momentum, as an iterative process by which an agent could attempt to sample from an unknown distribution $p(x)$ using only knowledge of its score function $\nabla \log p(x)$, which can be learned from noisy observations drawn from the distribution. The following points provide more detail on the sampling process and why Langevin dynamics arise:
> > * Biological networks such as those underlying navigation must accurately estimate environmental state variables such as position from observations of self-motion, even in the presence of intrinsic noise.
> > * This noise induces a distribution over network states, i.e., activity, however, it is unknown exactly what the true noise distribution is.
> > * Accurate state estimation would thus require the network to optimally remove intrinsic noise without access to the exact parameters of the noise distribution, but given noisy network states.
> > * This optimal solution can be accomplished by learning the score function of noisy network states and using it to denoise network states over time as additional inputs and noise are presented (Miyasawa, 1961).
> > * When the network receives no inputs, its noise-driven dynamics still attempt to remove intrinsic noise, but the learned score function is for the distribution of waking task-like activity. This causes the network states to resemble actual samples drawn from the distribution over network states in the presence of inputs. The theoretical results show that the network dynamics during this quiescent state carry out sampling by Langevin dynamics, i.e., using the score function of the network's activity distribution during task performance to yield quiescent network activity that resembles waking activity.
> >
> > > W1.2. Also, how can an RNN sleep? Do the authors refer to running RNNs without any inputs? If so, what are the initial conditions?
> >
> > Yes, the reviewer’s understanding is correct. We had mentioned that quiescence or sleep is the case when no inputs are provided to the RNN in Section 2.2 (currently Line 151-152, see black text “when no input is given”) and in Theorem 2 (currently Line 196-197). We have made this more explicitly clear in multiple locations: Line 80 in the Introduction, Line 120-121 in the Background, Section 2.1 and improvements in Section 2.2.
> >
> > We had provided details on initial conditions in Appendix A.1.2 & A.1.3 (see “Hidden state initialization” in the former and point 2 in the latter for the RatInABox task) and a reference in the Numerical Results section to Appendix A for implementation details. We have _made this more explicit_ and provided the following details in the new revision:
> >
> > We note here that replay trajectories are obtained by first randomly initializing the hidden state of the RNN, following which we collect activations for several unrolled timesteps in the absence of inputs but in the presence of noise. Further details are provided in Appendix A.1.
> >
> > To answer the question in more detail and as explained in Section A.1.2 for T-maze and Triangle tasks, RNN hidden states are initialized such that _initial readouts thereof start at the correct positions_, but have _additional task-dependent noise added to them that is orthogonal to the readout_. Without this task-dependent noise, learning to produce two different sets of trajectories that start from the same point in space becomes much more difficult in these tasks. For the RatInABox task described in Section A.1.3, _positions are randomly initialized_, but in such a way that initial positions are the _centroid of the top 3 most active place cells in a random place cell activation vector_ (sampled uniformly so as to tile space).

---

> ### Author Response · Authors · 2025-11-21
>
> > W2. … authors are rediscovering the fact that discretized RNNs, with discretization time being equal to the neural decay constant, are actually not a good model of continuous neural dynamics
>
> We agree that the reviewer's observation is a correct interpretation of Figure 2, however, it ignores or misses our (1) sampling-based perspective and (2) theoretical contributions that underlie the finding that leakage is important for replay.
>
> First, we would like to clarify that the key prior work that we develop upon, Krishna et al. [1], consider continuous-time RNNs, i.e., with “leakage” or properly set time constant and decay values such that $\Delta t << \tau$. Given that Levenstein et al. [2], another important work that explored the effects of adaptation/masked training, used discrete-time RNNs for their experiments, we wanted to clarify and bridge the gap between these two works and identify how important leakage or continuous-time dynamics actually are for both path-integration and replay.
>
> When we claim that leakage is important for replay, we do so not from a perspective of biological realism, but rather from a perspective of what dynamics an arbitrary system would need in order to perform Langevin sampling on some distribution of time-series.
>
> Thus, our claim is not simply that leakage is important for path-integration, as the reviewer suggests. Rather, we show “the score function is time-variant and difficult to estimate, even for simple distributions, but our expression of it motivates the addition of leakage (linear dynamics) in RNNs.” The first half of our claim, i.e., on the score function + specifically showing why leakage is useful to estimate it, is a key, non-trivial point that is missing from the reviewer's analysis. Our theoretical analysis of path-integration does not show that linearity is useful in addition to a non-linearity. Rather, we analytically show that linearity is sufficient for path-integrating any Gaussian process, but said linearity is a complicated function of time. Only then do we use our analysis as motivation to examine the role of leakage in training path-integrating networks, where it benefits networks trained with masking more than without.
>
> In summary, we approach the question of leakage not simply from the standpoint of being true to continuous-time dynamics but more specifically _motivate its usage because of its utility in estimating the score function_, which is _important not just for optimal path integration in the presence of noise but consequently for replay as well_.
>
> ---
>
> We hope that our comments, clarifications, and updates to the manuscript have addressed the reviewer’s concerns. If so, we would kindly request the reviewer to reconsider their assessment and potentially recommend acceptance. We are happy to address any other questions or concerns the reviewer may have. We thank the reviewer for their time and comments that we believe have improved our exposition and the paper’s accessibility.
>
>
> **References**
> 1. Krishna, Nanda H., et al. "Sufficient conditions for offline reactivation in recurrent neural networks." The Twelfth International Conference on Learning Representations.
> 2. Levenstein, Daniel, et al. "Sequential predictive learning is a unifying theory for hippocampal representation and replay." bioRxiv (2024): 2024-04.

---

> > ### Comment · Reviewer_WDVo · 2025-11-25
> >
> > I appreciated the author responses, thank you for the hard work. Admittedly, I was part of the broader audience that would read this paper but is not working immediately on the topic. Hence, it is not surprising that I missed several key contributions as listed by the authors.
> >
> > I will increase my score to 6, with the caveat that I cannot fully verify the originality of the claims with respect to the immediate prior works. I will leave it to more immediate experts to scrutinize the details of the focused claims, but this is a paper I would have liked to read. I will also lower my confidence to reflect this.

---

> ### Author Response · Authors · 2025-11-25
>
> We thank the reviewer for their acknowledgment of our rebuttal and recognizing our contributions. We are grateful that the reviewer considered increasing their score. We have attempted to summarize the state of the literature here to aid the reviewer in their evaluation:
>
> **Previous work:**
> - Continuous attractor networks (CANNs) can model path-integration (awake activity) and replay (internally driven activity) in networks like the hippocampus.
> - However, CANN models are not data-driven: they are hand-crafted, and are generally not learned from data.
> - Krishna et al. (2024) proposed a theory of replay generation (i.e., attractor learning) via path-integration and denoising that implicitly produce Langevin dynamics.
> - Levenstein et al. (2024) established that RNNs trained to path-integrate and denoise trajectories in complex environments behave similarly to CANNs.
>
> **Gaps in literature that we address:**
> 1. Theory from Krishna et al. (2024) assumed stationary activity, but in practice, both _in silico_ and _in vivo_, navigation trajectories are non-stationary. We identify what optimal dynamics may look like for non-stationary trajectories (Section 3.1, used in Figure 1).
> 2. Krishna et al. (2024) used "continuous" RNNs (i.e., leakage), while Levenstein et al. (2024) did not. We investigate what the role of leakage may be in path-integrating networks (Section 3.2).
> 3. Levenstein et al. (2024) and other (i.e., CANN-based) works have used adaptation (negative feedback) to induce exploration in replay trajectories. We show that the mechanism through which this happens (attractor destabilization) also slows replay (Sections 4.1 and 5).
> 4. To our knowledge, no existing works have proposed mechanisms to temporally compress activity (i.e., replay) from RNNs trained to path-integrate. We propose underdamped Langevin sampling via momentum (Section 4.2), and show that it can counter adaptation (Section 5, Figures 3 and 4) and speed up replay (Section 5, Figure 5). Crucially, we show that this temporal compression does not prevent adaptation-induced exploration (Section 5, Figures 6 and 7).

---

> ### Comment · Reviewer_WDVo · 2025-11-25
>
> I thank the authors for the explanation. To clarify my response, I am able to follow these arguments. I am just not an immediate expert to verify these arguments beyond these papers cited by the authors. I concur with their point 4, this is my assessment of the situation as well. I fully support acceptance, but cannot increase my score further only due to my lack of specific expertise. If I did, I would have!

---

> > ### Author Response · Authors · 2025-11-25
> >
> > We thank the reviewer for their positive comments and supporting acceptance, and for clarifying their stance.

---

### Official Review · Reviewer_MAde · 2025-11-01

**Soundness:** 3
**Presentation:** 3
**Contribution:** 2
**Rating:** 6
**Confidence:** 3

**Summary:**

This work studies the noisy recurrent neural networks (RNNs) from the perspective of biological neural networks (like the hippocampus). Since hippocampus generate the “replay” stimulus — internally-generated sequences of activity during quiescent periods (e.g., rest or sleep) that resemble trajectories during active behavior, many works have started to analyze noisy RNNs which mimic this behavior by path-integration. This work focuses on RNNs that implicitly learn to acts as generative models over a fixed distribution of input sequences. More specifically, how noisy RNNs that are trained to path-integrate their inputs implicitly learn statistics that produce Langevin sampling of the input distribution when no input is given. Earlier works have assumed that the distribution of RNN activity p(r(t)) is stationary and thus the score function depends only on r(t). This work refutes this assumption and argues in favor of the role of hidden state leakage in path-integration. This work shows that hidden state adaptation (negative feedback) allows for exploration in replay, but results in non-Markov sampling along with slow down in replay. Further, it proposes model for temporally compressed replay in noisy path-integrating RNNs through hidden state momentum term resulting in underdamped Langevin sampling. Numerical experiments are conducted for path-integration to verify the proposed model.

**Strengths:**

- This works shows that path-integration in noisy RNNs results in score functions which are time-variant and difficult to estimate even for simple distributions such as Gaussian. It demonstrates a useful inductive bias of hidden state linear leakage.
- This paper shows that Adaptation (negative feedback) induces exploration in replay. It results in non-Markov second-order Langevin sampling destabilizing attractors, results in diversification/exploration in replay as well as slowing down replay process.
- The proposed method of adding momentum to induce underdamped Langevin sampling induces fast replay resulting in temporal compression replay and overcomes replay slow-down induced by adaptation while still maintaining replay diversity/exploration.

**Weaknesses:**

- It is unclear if the proposed under-dampening mechanism with momentum is biologically plausible?
- The theoretical analysis has been done under the Gaussian assumption, it’s not clear if the same would hold true when this assumption breaks down.
- While the proposed work focuses on the effects of two counteracting terms : adaptation (negative feedback) and underdampening (momentum). Since adaptation slows replay along with adding  diversity/exploration in the replay. In contrast, under-dampening accelerates replay due to momentum term. These two terms are at odds with each other and while carefully balancing helps solve some of the easy tasks presented in this work, it’s unclear what a systematic approach would encourage. It would have been better to explore this aspect in a bit more detail in this work.

**Questions:**

- In Fig, 2, any clues as to why leakage is more helpful for T-maze problem compared to Triangle? Since it’s clear with increasing the masking difficulty (k), the convergence is better for T-maze with leakage but there’s less gap in Triangle setup.
- What is the behavior of the experiment in Fig.2 without masked training ? Does the leakage term still helps in this case?
- How biologically plausible is the proposed under-dampening mechanism of added momentum term $(1-\lambda_v) v(t)$ ?
- How does one extend the proposed insights from path-integration tasks to richer tasks like planning and policy learning?
- In Fig. 4, underdampening constant $\lambda_v$ is greater than 0.5, what are the effects when this constant goes down?
- Although most of the experimental setup uses ReLU RNNs, would the empirical analysis change drastically in the presence of Gated RNNs or RNNs with other non-linear activation functions?

---

> ### Author Response · Authors · 2025-11-21
>
> We would like to thank the reviewer for their detailed review and questions, and for their overall positive assessment of our work. We respond to the reviewer’s concerns and questions below, and also in the [response to common questions](https://openreview.net/forum?id=kGNtM8NteM&noteId=SRBMqMYQpI).
>
> > W1. It is unclear if the proposed under-dampening mechanism with momentum is biologically plausible?
> >
> > Q3. How biologically plausible is the proposed under-dampening mechanism … ?
>
> We thank the reviewer for these questions and request them to refer to our [response to common questions](https://openreview.net/forum?id=kGNtM8NteM&noteId=SRBMqMYQpI). To summarize, several prior studies have theoretically examined methods for fast sampling in the brain, incorporating mechanisms like momentum, and recent experimental evidence suggests that momentum effects can be observed in human random sampling as well as rat hippocampal sharp wave ripples. While we leave a more detailed biological mapping of such mechanisms (e.g., onto E/I circuits) to future work, we argue that the brain must be using techniques for faster sampling, such as momentum, from a speed and efficiency perspective as discussed in prior work [1,2].
>
> > W2. The theoretical analysis has been done under the Gaussian assumption, it’s not clear if the same would hold true when this assumption breaks down.
>
> We thank the reviewer for this important question. We provide arguments below justifying our assumptions and explaining how they would extend to non-Gaussian cases:
> * First, even if noise in neural circuits is non-Gaussian, we argue that by the central limit theorem, independent noise to each neuron would mean that the noise distribution for the whole population would be Gaussian (especially as the population sizes get larger).
> * Specifically considering Poisson noise if interpreting RNN activations as firing rates, we argue that the Poisson distribution approaches a Gaussian distribution as its parameter $\lambda$ approaches larger values (i.e., as $\lambda \xrightarrow{} \infty$). Thus, the Gaussian noise assumption would be reasonable.
> * Furthermore, the Tweedie-Miyasawa formula used here as the optimal solution to denoising applies not only to Gaussian distributions but extends to the general class of exponential family distributions [3,4] (e.g., Poisson, gamma, beta, binomial, multinomial, etc.; see also Masset et al. [2] which describes a general framework for Langevin sampling and modifications, although focused on the Gaussian case, but readily applicable to non-Gaussian scenarios).
> * Finally, we refer to several prior works in theoretical neuroscience with similar assumptions on neuronal noise [1,2,5–7] and argue that our model is consistent with traditional work in the field of theoretical neuroscience and machine learning.
>
> Overall, even if one considers denoising from non-Gaussian or deterministic perturbations, a plethora of works make it abundantly clear that generative sampling via diffusive processes (i.e., implicitly estimating the score function of input data) is still viable [8–12]. In general, learning the score function is important for generative sampling.
>
> We would also like to note that in our experiments, the noise statistics are isotropic Gaussian. Thus, the Gaussianity assumption for the theoretical result is not broken in our work.

---

> > ### Author Response · Authors · 2025-11-21
> >
> > > W3. [Adaptation and under-dampening] are at odds with each other and while carefully balancing helps solve some of the easy tasks presented in this work, it’s unclear what a systematic approach would encourage. It would have been better to explore this aspect in a bit more detail in this work.
> >
> > We apologize, but it is not fully clear to us what the reviewer had in mind with this comment. We respond to it below based on our understanding, and hope it clarifies the reviewer’s concern. If the reviewer has other concerns, we are happy to address them in the discussion phase.
> >
> > Firstly, the _sole task_ of the RNN models is to _path-integrate velocities to estimate position_, as one would for navigation; this is done _without any second-order modifiers like adaptation or underdampening/momentum_. Replay trajectories are simply byproducts of path-integration training, and _second-order modifiers are only introduced at inference_ (i.e., replay generation) time. In other words, the _task the RNNs are trained to do is independent of adaptation and underdampening_.
> >
> > Secondly, continuous attractor dynamics underlie several fundamental computations implemented by neural circuits apart from navigation, such as evidence integration (line attractor) and working memory (ring attractor). An optimal solution to these tasks is path-integration, so our results are generalizable to these different task scenarios as well [13].
> >
> > Finally, if the reviewer is referring to replay generation as the "task", then it must be acknowledged that encouraging exploration and temporal compression in replay is non-trivial; while others have explored the former via adaptation, to date there are very few models for the latter, so we would not describe temporally compressing replay as an "easy" task. Additionally, we do make sure to thoroughly compare combinations of adaptation and underdampening with respect to replay speed and exploration: while there is a clear tradeoff in the former, we are the first to show, counterintuitively, that there is on average no such tradeoff in the latter.
> >
> > > Q1. In Fig. 2 … why leakage is more helpful for T-maze problem compared to Triangle?
> >
> > We thank the reviewer for the insightful question. We believe that the T-maze task training is more sensitive to leakage than the triangle task training because T-maze trajectories are less predictable: the first half of their trajectories are all identical, until they reach a sudden bifurcation before moving in opposite directions; meanwhile, triangle paths are always straight and only share their starting points.
> >
> > > Q2. What is the behavior of the experiment in Fig.2 without masked training? Does the leakage term still helps in this case?
> >
> > We apologize, but we believe there may be a misunderstanding here. Masked training uses $k > 1$; meanwhile _when $k=1$, training is exactly unmasked training_, since the RNN observes the true velocity of a trajectory at every timestep (see Equation 14). The majority of Figure 2 examines training during the $k=1$ (unmasked) regime. In this regime, leakage still helps, but is not as important as in the masked $k > 1$ regimes. Thus, to summarize, Figure 2 already shows unmasked training, i.e., $k=1$, for most training steps, where we see that leakage is at least slightly useful, especially in the more sensitive training process for T-maze (as answered above). We hope this clarifies things for the reviewer and are happy to address any outstanding concerns here.

---

> > > ### Author Response · Authors · 2025-11-21
> > >
> > > > Q4. How does one extend the proposed insights from path-integration tasks to richer tasks like planning and policy learning?
> > >
> > > We thank the reviewer for the great question. Our theoretical results hold for networks that are trained on tasks involving some path-integration, e.g., variants of navigation as considered here, head direction estimation, working memory tasks, evidence integration, etc. which all use continuous attractor dynamics. Thus, our results may be _easily extended to all these fundamental tasks_ that must be performed by neural circuits.
> > >
> > > Planning and policy learning are closely related to these tasks and are carried out by regions downstream of those such as the hippocampus, e.g., the frontal lobe and the neocortex. Firstly, networks trained to navigate or more generally path-integrate learn a cognitive map of the environment [14]. This computation is essential for any system that must navigate in a real-world environment. Furthermore, _sampling plausible trajectories using this learned cognitive map could be critical for planning_ or learning in environments where training experiences are sparse. Recent work [15] has exactly discussed the former, showing that hippocampal-entorhinal spatial codes are used in conjunction with frontal lobe mechanisms to plan routes during navigation. Thus, _trajectories generated by noise-driven dynamics_ as shown in our work could also be used to _sample proposed paths for navigation plans_ [16,17]. Furthermore, replayed activity from circuits such as those we study could serve as an input to a separate memory network forming neuronal assemblies through synaptic plasticity (e.g., a Hopfield network). This extra input data to such downstream networks, such as the neocortex (which is implemented in decision making) could facilitate improved consolidation. This is in line with the complementary learning systems theory [18].
> > >
> > > > Q5. In Fig. 4, … what are the effects when [friction $\lambda$] goes down [(at or below 0.5)]?
> > >
> > > We thank the reviewer for this question. Decreasing friction $\lambda$ increases the oscillatory behavior of replay trajectories, but also does make them travel farther. Moreover, having $\lambda$ too low (i.e., below 0.5 in most experiments) can incur stability issues. This is to be expected from gradient ascent with momentum (which underdamped Langevin sampling is). We generally observed $\lambda$ greater than or equal to 0.7 yielded the best combination of stability, speed, and minimal aberrations.
> > >
> > > > Q6. … would the empirical analysis change drastically in the presence of Gated RNNs or RNNs with other non-linear activation functions?
> > >
> > > We thank the reviewer for this important question. We have provided a response to this question in our [response to common questions](https://openreview.net/forum?id=kGNtM8NteM&noteId=oGReodDPn1). In short, the overall results do not change drastically and remain largely consistent in the presence of other activation functions such as $\mathrm{tanh}$, as explained by our new results in Appendix G.
> > >
> > > ---
> > >
> > > Overall, we thank the reviewer again for their positive review and for engaging with our work. We hope that our response clarifies the reviewer’s concerns, and kindly request the reviewer to consider supporting our work more strongly if so. We are also happy to address any further concerns from the reviewer during the discussion phase.

---

> > > > ### Author Response · Authors · 2025-11-21
> > > >
> > > > **References**
> > > >
> > > > 1. Hennequin, Guillaume, Laurence Aitchison, and Máté Lengyel. "Fast sampling-based inference in balanced neuronal networks." Advances in neural information processing systems 27 (2014).
> > > > 2. Masset, Paul, et al. "Natural gradient enables fast sampling in spiking neural networks." Advances in neural information processing systems 35 (2022): 22018-22034.
> > > > 3. Efron, Bradley. "Tweedie’s formula and selection bias." Journal of the American Statistical Association 106.496 (2011): 1602-1614.
> > > > 4. Kim, Kwanyoung, and Jong Chul Ye. "Noise2score: tweedie’s approach to self-supervised image denoising without clean images." Advances in Neural Information Processing Systems34 (2021): 864-874.
> > > > 5. Bredenberg, Colin, Eero Simoncelli, and Cristina Savin. "Learning efficient task-dependent representations with synaptic plasticity." Advances in neural information processing systems 33 (2020): 15714-15724.
> > > > 6. Bredenberg, Colin, et al. "Impression learning: Online representation learning with synaptic plasticity." Advances in neural information processing systems 34 (2021): 11717-11729.
> > > > 7. Bittner, Sean R., et al. "Interrogating theoretical models of neural computation with emergent property inference." Elife 10 (2021): e56265.
> > > > 8. Daras, Giannis, et al. "Soft Diffusion: Score Matching with General Corruptions." Transactions on Machine Learning Research (2023).
> > > > 9. Huang, Xingchang, et al. "Blue noise for diffusion models." ACM SIGGRAPH 2024 conference papers. 2024.
> > > > 10. Nachmani, Eliya, Robin San Roman, and Lior Wolf. "Denoising diffusion gamma models." arXiv preprint arXiv:2110.05948 (2021).
> > > > 11. Rissanen, Severi, Markus Heinonen, and Arno Solin. "Generative Modelling with Inverse Heat Dissipation." International Conference on Learning Representations. 2023.
> > > > 12. Zhou, Mingyuan, et al. "Beta diffusion." Advances in Neural Information Processing Systems 36 (2023): 30070-30095.
> > > > 13. Krishna, Nanda H., et al. "Sufficient conditions for offline reactivation in recurrent neural networks." The Twelfth International Conference on Learning Representations.
> > > > 14. Whittington, James CR, et al. "How to build a cognitive map." Nature neuroscience 25.10 (2022): 1257-1272.
> > > > 15. Epstein, Russell A., et al. "The cognitive map in humans: spatial navigation and beyond." Nature neuroscience 20.11 (2017): 1504-1513.
> > > > 16. Kay, Kenneth, et al. "Constant sub-second cycling between representations of possible futures in the hippocampus." Cell 180.3 (2020): 552-567.
> > > > 17. Wang, Jiyi, et al. "Brain-Like Replay Naturally Emerges in Reinforcement Learning Agents." arXiv preprint arXiv:2402.01467 (2024).
> > > > 18. McClelland, James L., Bruce L. McNaughton, and Randall C. O'Reilly. "Why there are complementary learning systems in the hippocampus and neocortex: insights from the successes and failures of connectionist models of learning and memory." Psychological review 102.3 (1995): 419.

---

### Author Response · Authors · 2025-11-21
**Response to common questions**

We thank the reviewers for engaging with our work and offering valuable comments. We note here that updates to the text in our manuscript are colored in forest green. Key changes include the addition of background details in line with reviewer suggestions, which we will mention in reviewer-specific comments, and additional results with $\mathrm{tanh}$ networks. Below, we provide responses to some common questions from the reviewers.

**Biological plausibility of underdamped Langevin sampling via momentum**

Some reviewers have understandably questioned the biological plausibility of our underdamped Langevin sampling in RNNs via hidden state momentum. In general, any system that attempts to sample fast and efficiently should make use of modifiers such as momentum as opposed to standard Langevin sampling, which is critically slow (i.e., mixing times are very long and convergence to the target steady-state distribution is very slow) [1].

Several previous works in theoretical neuroscience have proposed mechanisms involving momentum, including using Hamiltonian dynamics, for fast and improved sampling [1–6]. Some of these have also mapped the implementation of such dynamics to more detailed E/I networks and spiking networks, which future work developing upon ours could consider. Thus, it is conceivable that the brain could be using such sophisticated sampling schemes involving momentum as it must operate quickly and efficiently [1]. Furthermore, experimental evidence indicates that a sampling scheme with momentum accounts for human random sequence generation better than one without [7], and hippocampal sharp wave ripples have been shown to simulate trajectories with momentum [8].

Thus, we argue that momentum in sampling could reasonably be implemented by circuits in the brain. As Krishna et al. [9] is the first theoretical work to posit that replay could be viewed as sampling in trained/learned recurrent networks, we attempt to better account for the temporal compression of such replay sequences using momentum.

**On the plausibility of momentum for temporal compression of replay sequences**

The underlying physiological mechanisms behind temporal compression are not yet fully understood, so attempts to map theoretical circuits that compress sequences onto biological neurons would be difficult and outside the scope of our work. Nonetheless, here we outline a series of arguments that together suggest momentum may be a reasonable initial approximation of neural processes, whatever they may be, that underlie replay generation and temporal compression:
* If a biological system were to implement sampling based on log-likelihood gradient ascent, then momentum would be a useful inductive bias towards this end, as it is in many other gradient-based optimization procedures [1–6].
* Hippocampal sharp wave ripple trajectories have been observed to contain momentum [8].
* Short-term facilitation can induce phase precession and temporal compression within spiking networks [10]. However, extensions to firing-rate networks like ours are scant, so it is certainly possible that momentum can act as a reasonable rate-approximation of short-term facilitation in spiking networks.
* Short-term post-synaptic plasticity (facilitation) via NMDA receptors has been shown to induce momentum-like effects in continuous attractor networks that track stimuli (Figures 3 and 4 of Zhao et al. [11]).

Here, we also argue that our theoretical work has value in neuroscience [12], where biological experiments can be expensive and difficult. Our work should be seen as understanding key components of and improving upon existing theoretical results to account for experimental observations in replay, such as temporal compression and exploration. Lying at the intersection of mechanistic and normative modeling (i.e., explaining how and why a phenomenon might arise), our theoretical work proposes mechanisms that could better account for replay in biological circuits, and also outlines paths for future experiments on replay, such as probing the connection between short-term facilitation and temporal compression of replay events.

---

> ### Author Response · Authors · 2025-11-21
> **Response to common questions (contd.)**
>
> **Investigation of claims with non-ReLU RNNs**
>
> Some reviewers have understandably asked whether our claims hold in non-ReLU RNNs. To this end, we ran our experiments with trained $\mathrm{tanh}$ RNNs. We see that _momentum absolutely still induces temporal compression, and adaptation induces temporal dilation, in replay from tanh RNNs_ (like in Figure 5). Both mechanisms still generally increase Wasserstein distances from awake trajectories (like in Figure 4). As for exploration, saturating (e.g., $\mathrm{tanh}$) RNNs have less stochastic replay trajectories, and explore less than their ReLU RNNs, so the effects of underdampening on exploration in $\mathrm{tanh}$ RNNs are mixed. We have added these results to our updated manuscript in Appendix G.
>
> In general, we argue that ReLU RNNs offer better biological realism leading to their prevalence in all prior work we have considered, as outlined in Appendix A.2.1.  Compared to RNNs with multiplicative gating interactions (e.g., GRUs and LSTMs) or activations such as $\mathrm{tanh}$, ReLU RNNs are more biologically plausible insofar as they can be easily interpreted as the nonnegative, sparse firing rates of spiking networks. This is a key reason (among others listed in Appendix A.2.1) that motivates the use of ReLU networks in our work.
>
> **On quantitative evidence**
>
> While some reviewers have mentioned that our claims are supported primarily by qualitative evidence, we would like to respectfully disagree on this point and hope that our comprehensive experimental results substantiate this. While we do not compare to biological data, we have provided several quantitative metrics to support our claims and theory in our numerical results, and we argue that given the difficulty of experiments on replay, our theoretical results should be seen as providing directions for experimentalists to pursue. Overall:
> * Comparing neural data and artificial network activations is non-trivial. This is especially the case with replay. Replay events are sparse and embedded in a sea of complex, noisy dynamics. Detecting replay events in vivo is itself a research challenge, so comparing with replay data would be outside the scope of our work, especially considering that there would be no clear mapping from an artificial neural network to a biological one. Future work could evaluate our predictions in biological networks through targeted experiments. We further direct reviewers to a previous response on the value of theoretical modeling in neuroscience (under “On the plausibility of momentum for temporal compression of replay sequences”).
> * We try to address a lack of biological data by working with synthetic models that mimic it, such as RatInABox [13]. We simulated synthetic rat trajectories and place cells, then trained an RNN to path-integrate place cell activity, and then decoded place cell activity during replay to analyze 2D synthetic rat replay trajectories (Figures 4–11).
> * We quantified several characteristics of replay trajectories and compared them numerically to those of awake trajectories: in Figure 4, we compute Wasserstein distances (a very common metric for generative models, which our RNNs are); in Figures 5 and 9, we estimate the summary statistics of the speed of replay; in Figure 6, we calculate the average length of replay trajectories; in Figures 7 and 12, we calculate numerical measures of exploration (note that exploration has no existing standard metrics); and in Figures 10 and 11, we calculate average mean displacements (like McNamee et al. [14]) and variances of replay trajectories over time.
>
> We thank the reviewers again and hope that our general and specific comments address the reviewers’ concerns. We look forward to discussing further with the reviewers and request them to reconsider their assessment based on our rebuttal.

---

> > ### Author Response · Authors · 2025-11-21
> > **Response to common questions (contd.)**
> >
> > **References**
> >
> > 1. Hennequin, Guillaume, Laurence Aitchison, and Máté Lengyel. "Fast sampling-based inference in balanced neuronal networks." Advances in neural information processing systems27 (2014).
> > 2. Masset, Paul, et al. "Natural gradient enables fast sampling in spiking neural networks." Advances in neural information processing systems 35 (2022): 22018-22034.
> > 3. Furlong, P. Michael, et al. "Biologically-Plausible Markov Chain Monte Carlo Sampling from Vector Symbolic Algebra-Encoded Distributions." International Conference on Artificial Neural Networks. Cham: Springer Nature Switzerland, 2024.
> > 4. Dong, Xingsi, and Si Wu. "Neural sampling in hierarchical exponential-family energy-based models." Advances in Neural Information Processing Systems 36 (2023): 78593-78606.
> > 5. Aitchison, Laurence, and Máté Lengyel. "The Hamiltonian brain: Efficient probabilistic inference with excitatory-inhibitory neural circuit dynamics." PLoS computational biology 12.12 (2016): e1005186.
> > 6. Dong, Xingsi, et al. "Adaptation accelerating sampling-based bayesian inference in attractor neural networks." Advances in Neural Information Processing Systems 35 (2022): 21534-21547.
> > 7. Castillo, Lucas, et al. "Explaining the flaws in human random generation as local sampling with momentum." PLOS Computational Biology 20.1 (2024): e1011739.
> > 8. Krause, Emma L., and Jan Drugowitsch. "A large majority of awake hippocampal sharp-wave ripples feature spatial trajectories with momentum." Neuron 110.4 (2022): 722-733.
> > 9. Krishna, Nanda H., et al. "Sufficient conditions for offline reactivation in recurrent neural networks." The Twelfth International Conference on Learning Representations.
> > 10. Leibold, Christian, et al. "Temporal compression mediated by short-term synaptic plasticity." Proceedings of the National Academy of Sciences 105.11 (2008): 4417-4422.
> > 11. Zhao, Huilin, Sungchil Yang, and Chi Chung Alan Fung. "Short-term postsynaptic plasticity facilitates predictive tracking in continuous attractors." Frontiers in Computational Neuroscience 17 (2023): 1231924.
> > 12. Levenstein, Daniel, et al. "On the role of theory and modeling in neuroscience." Journal of Neuroscience 43.7 (2023): 1074-1088.
> > 13. George, Tom M., et al. "RatInABox, a toolkit for modelling locomotion and neuronal activity in continuous environments." Elife 13 (2024): e85274.
> > 14. McNamee, Daniel C., et al. "Flexible modulation of sequence generation in the entorhinal–hippocampal system." Nature neuroscience 24.6 (2021): 851-862.

---

### Author Response · Authors · 2025-12-01

In this comment, we would like to summarize our rebuttal, to aid the new Area Chair in evaluating our work. We provide a list of key concerns from the reviewers and a summary of our response in each case. For responses to other questions and greater detail, we request the Area Chair and readers to refer to our comments responding to the reviewers.

**Reviewer MAde (Rating: 6, Confidence: 3)**

|Concern|Response|
|-|-|
|Biological plausibility of underdampening via momentum|If biological networks implement sampling, then momentum is natural for sampling efficiency/speed. Other neuroscientific works have proposed momentum-like Hamiltonian sampling dynamics, and have shown explicit momentum or momentum-like behavior both in vivo and in silico. See "Response to common questions".|
|Whether Langevin sampling theory holds in non-Gaussian settings|Gaussianity is a common and reasonable assumption in theoretical neuroscience, we argue based on: (i) the central limit theorem; (ii) the key result is generally applicable to exponential family distributions; and (iii) a plethora of works have shown the viability of denoising-based sampling in non-Gaussian settings.|
|Confusion about task and combinations of underdampening + adaptation|We clarify that (a) our RNNs are only trained to path-integrate, not to generate replay, and (b) we tried several combinations of underdampening and adaptation.|
|Experiments with non-ReLU RNNs|We performed experiments with tanh RNNs (Appendix G). The general observations with ReLU RNNs hold: underdampening/adaptation temporally compresses/dilates replay. Trends in exploration were more mixed because tanh RNNs explored less than ReLU RNNs. See “Response to common questions”.|

**Reviewer WDVo (Rating: 4, Confidence: 3 → Rating: 6, Confidence: 2)**

|Concern|Response|
|-|-|
|Confusion about background|We clarified relevant background, expanded our Background portion of the manuscript, and added an additional Background overview in Appendix B.|
|Importance of leakage|We clarified that leakage is not just important because it is necessary for continuous-time dynamics, but more importantly because it is necessary to learn the score function for optimal path integration in the presence of noise.|

The reviewer responded that they *fully supported acceptance*, and could not increase their score further only due to their lack of specific expertise.

**Reviewer Qmcp (Rating: 6, Confidence: 4)**

|Concern|Response|
|-|-|
|Biological plausibility of underdampening via momentum|See response to MAde Concern 1.|
|Purported lack of quantitative evidence|We clarified that (a) we have several quantitative metrics (Wasserstein distance in Figure 4, speed in Figure 5, and two measures of exploration in Figures 6 and 7), and (b) comparison with biological replay is very difficult, but our synthetic rat place cell tasks try to approximate this. See “Response to common questions”.|
|Experiments with non-ReLU RNNs|See response to MAde Concern 4.|

**Reviewer FE8C (Rating: 4, Confidence: 2)**

|Concern|Response|
|-|-|
|Motivation of adaptation and underdampening mechanisms|We clarify that our choice of adaptation and underdampening mechanisms are grounded in previous works and Langevin sampling theory.|
|Whether our work is about modeling biological circuits + neural data or finding ways to improve RNN training|We clarify that our work is focused on theoretical neuroscience and path-integrating RNNs, not machine learning, and that direct comparison with biological replay is very difficult.|
|Confusion about how the theoretical results in Section 2 (Background) support our claims|We clarify how background content relates to our work, and added a Background overview in Appendix B.|
|Purported lack of quantitative data or biological experiments|See response to Qmcp Concern 2.|

We thank the reviewers for their time and comments, and hope that our responses have alleviated their concerns. We also hope that the Area Chair finds our responses convincing and this summary useful in their decision making.

---

### Meta-Review · Area_Chair_gUGK · 2026-01-06

**Summary:**

Reviewers found the setting and theoretical direction interesting, but raised concerns about clarity, positioning, biological interpretation, and the strength of evidence supporting the main claims. These concerns informed the decision.

**Reviewer Concerns:**

The rebuttal addressed several issues, including added background, clarifications, additional experiments, and quantitative metrics. However, key concerns remain: the biological link of the discussed mechanisms remains largely speculative, and the scope and positioning between neuroscience and machine learning are still unclear.

**Reviewer Scores:**

Reviewer scores would likely remain largely unchanged. While one reviewer increased their score after the rebuttal, they explicitly reduced confidence. Reviewers who were below the acceptance threshold would likely have remained so, leaving the overall assessment borderline and below the acceptance bar.

---

### Decision · Program_Chairs · 2026-01-26

Reject